# Multi-resolution localization of causal variants across the genome

Matteo Sesia [1], Eugene Katsevich [1], Stephen Bates [1], Emmanuel Candès[2✉] & Chiara Sabatti[3✉]

In the statistical analysis of genome-wide association data, it is challenging to precisely localize the variants that affect complex traits, due to linkage disequilibrium, and to maximize power while limiting spurious findings. Here we report on *KnockoffZoom*: a flexible method that localizes causal variants at multiple resolutions by testing the conditional associations of genetic segments of decreasing width, while provably controlling the false discovery rate. Our method utilizes artificial genotypes as negative controls and is equally valid for quantitative and binary phenotypes, without requiring any assumptions about their genetic architectures. Instead, we rely on well-established genetic models of linkage disequilibrium. We demonstrate that our method can detect more associations than mixed effects models and achieve fine-mapping precision, at comparable computational cost. Lastly, we apply *KnockoffZoom* to data from 350k subjects in the UK Biobank and report many new findings.

[1] Department of Statistics, Carnegie Mellon University, Pittsburgh, PA, USA. [2] Departments of Mathematics and of Statistics, Stanford University, Stanford, CA 94305, USA. [3] Departments of Biomedical Data Science and of Statistics, Stanford University, Stanford, CA 94305, USA. ✉email: candes@stanford.edu; sabatti@stanford.edu

Since the sequencing of the human genome, there have been massive efforts to identify genetic variants that affect phenotypes of medical relevance. By genotyping single-nucleotide polymorphisms (SNPs) in large genome-wide association studies (GWAS), thousands of associations with different traits and diseases have been discovered[1]. However, it has been challenging to translate these findings into actionable knowledge[2]. As a step in this direction, we present a new statistical method that improves our ability to resolve the location of causal variants.

The analysis of GWAS data started by testing SNP-by-SNP hypotheses of marginal independence with the trait (not accounting for the rest of the genome) using univariate regression. Today, the leading approaches employ a linear mixed model (LMM) that includes a random term approximating the effects of other variants in distant loci[3–8]. Nonetheless, this still tests marginal independence because it does not account for linkage disequilibrium (LD)[9,10]. Thus, it cannot distinguish causal SNPs from uninteresting nearby variants, since neither are independent of the phenotype. This limitation becomes concerning as we focus on polygenic traits and rely on larger samples; in this setting, it has been observed that most variants are correlated with the phenotype, although only a fraction of them may be important[11]. Therefore, most null hypotheses of no association should be rejected[2], which is a rather uninformative conclusion. The awareness that marginal testing is insufficient has led to the heuristic practice of post hoc aggregation (clumping) of associated loci in LD[8,12], and to the development of fine mapping[13]. Fine-mapping methods refine marginally significant loci and discard associated but noncausal SNPs by accounting for LD, often within a Bayesian perspective[14–17]. However, this two-step approach switches models and assumptions in the middle of the analysis, obfuscating the interpretation of the findings and possibly invalidating type-I error guarantees. Moreover, as LD makes more noncausal variants appear marginally associated with the trait in larger samples, fine-mapping tools face an increasingly complicated task refining wider regions.

An alternative solution is suggested by recent advances in statistics, notably *knockoffs*[18], which can account for LD genome wide without heuristic post processing, while controlling the false-discovery rate (FDR)[19]. The only assumption of knockoffs is that LD is adequately described by hidden Markov models (HMMs) that have been successfully employed in many areas of genetics[20–23]. Moreover, since knockoffs do not require any model linking genotypes to phenotypes, they seamlessly apply to both quantitative and qualitative traits. In fact, the general validity of knockoffs for GWAS has been explored and discussed before[24–29].

Here we present *KnockoffZoom*: a new method that leverages knockoffs to address the current difficulties in locus discovery and fine mapping. *KnockoffZoom* searches for causal variants across the genome and reports SNPs (or groups thereof) that distinctly influence the trait accounting for the effects of all others. This is carried out by testing the conditional association of predefined groups of SNPs at multiple resolutions, ranging from that of locus discovery to that of fine mapping, while probably controlling the FDR. This work involves some key innovations. First, we develop algorithms to analyze the data at multiple levels of resolution, in order to maximize power in the presence of LD without pruning the variants[18,24]. Second, we improve the computational efficiency and apply our method to a large dataset, the UK Biobank[30]—a previously unfeasible task.

## Results

**KnockoffZoom.** In the marginal analysis, a variant is null if its allele distribution is independent of the phenotype. By contrast,

*KnockoffZoom* tests stricter conditional hypotheses: a variant is null if it is independent of the trait conditionally on all other variants, including its neighbors (Methods). Suppose that we believed a multivariate linear model realistically describes the inheritance of the trait[31], then, a conditional hypothesis would be non-null if the corresponding variant had a nonzero coefficient. Generally, in the absence of unmeasured confounders or feedback effects of the phenotype onto the genome, our tests lead to the discovery of causal variants. In particular, we can separate markers that have a distinct effect on the phenotype from those whose associations are merely due to LD.

LD makes conditional null hypotheses challenging to reject, for the same reason why colinearity reduces the power of $t$ tests in multivariate linear regression. In that case, $F$ tests may be preferable. Analogously, we group variants that are too similar for their distinct effects to be discerned, and test whether the trait is independent of all of them, conditional on the other groups. A group is null if it only contains SNPs that are independent of the phenotype, conditional on the others. Concretely, we define contiguous blocks at multiple resolutions by partitioning the genome via adjacency-constrained hierarchical clustering[32]. We adopt the $r^2$ computed by PLINK[12] as a similarity measure for the SNPs, and cut the dendrogram at different heights, see Fig. 1b. However, different choices are compatible with *KnockoffZoom*, as long as the groups are determined before looking at the phenotype data. In particular, contiguity is not necessary; we have adopted it because it corresponds to the idea that researchers want to localize causal variants within a genomic segment, and because it produces interpretable results when some of the genetic variation is not genotyped. In addition, contiguity tends to yield higher power when we assume an HMM for the distribution of the genotypes (Supplementary Methods).

To balance power and resolution, we consider increasingly refined partitions (Supplementary Table 1, Supplementary Fig. 1). Figure 1a shows an example of our results. Each rectangle in the Chicago plot (named for its stylistic resemblance to the Willis tower) spans a region that includes variants carrying distinct information about the trait compared with the other blocks on the same level. Blocks at higher resolutions (higher levels) are narrower and harder to discover.

We can mathematically prove that *KnockoffZoom* controls the FDR below any desired level $q$ (at most a fraction $q$ of our findings are false positives on average), at each resolution. The FDR is a meaningful error rate for complex traits[33,34], but its control is challenging[35]. We overcome this difficulty with *knockoffs*. These are carefully engineered synthetic variables that can act as negative controls because they are exchangeable with the genotypes and reproduce the spurious associations that we want to winnow[18,24,36,37]. Constructing knockoffs requires specifying the distribution of the genotypes, which we approximate as an HMM. In this paper, we implement the fastPHASE[21] HMM (Supplementary Methods, Supplementary Note 1), which we show works well for relatively homogeneous individuals (Supplementary Note 2, Supplementary Fig. 2), although it has some limitations. In particular, it does not describe population structure, and it is less accurate for rare variants (Supplementary Fig. 3). However, *KnockoffZoom* can easily accommodate other HMMs in the future. Meanwhile, we extend an earlier knockoff generation algorithm[24] to target multi-resolution hypotheses, and we reduce the complexity of this operation analytically (Supplementary Methods). With the UK Biobank data, for which the haplotypes have been phased[23], we accelerate this algorithm further by avoiding implicit re-phasing (Supplementary Methods).

To powerfully separate causal variants from spurious associations, we fit a multivariate predictive model of the trait, and

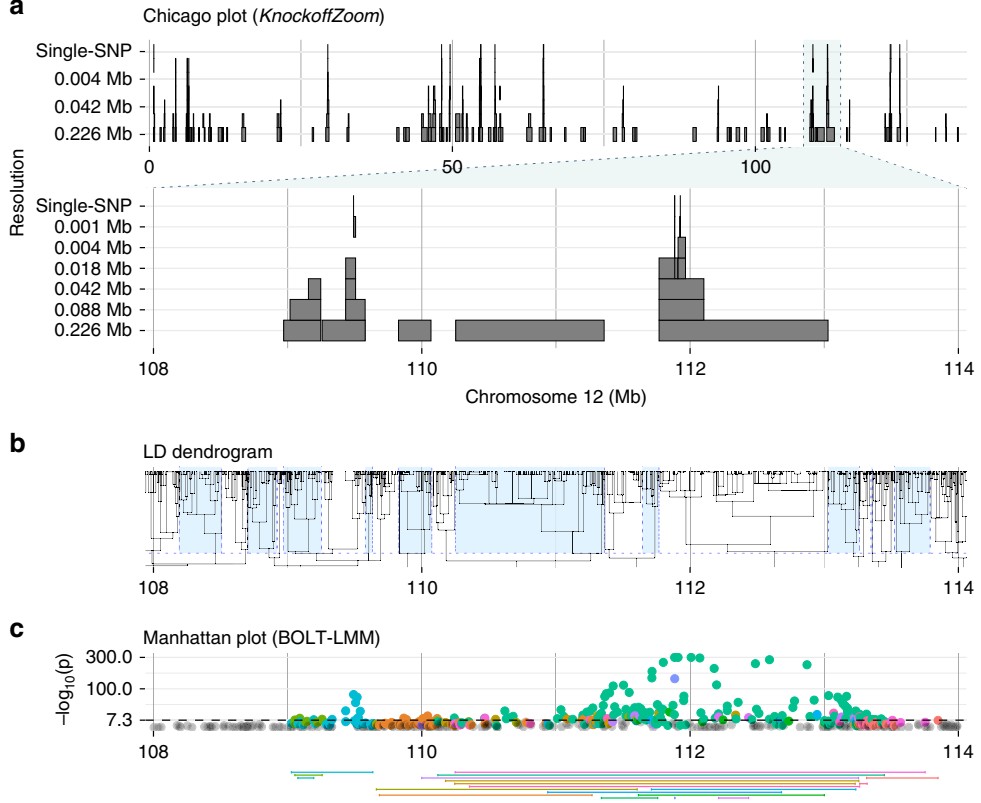

**Fig. 1 Discoveries for platelet count.** *KnockoffZoom* discoveries **a** on chromosome 12 for the phenotype platelet in the UK Biobank, controlling the FDR below 0.1. Each shaded rectangle represents a discovery at the resolution indicated by its vertical position (measured by the average width of the blocks), so that the highest-resolution findings are on top. The lower part of **a** focuses on a smaller genomic region. The hypotheses are prespecified by cutting the LD dendrogram **b** at different heights. As an example, by alternating blue and white shading in (**b**), we indicate the lowest-resolution blocks. The Manhattan plot **c** shows the BOLT-LMM *p* values from the same data, while the segments below represent the region spanned by the significant discoveries clumped with PLINK at the genome-wide significance level ($5 \times 10^{-8}$). For each clump, the colors match those of the corresponding *p* values. All plots are vertically aligned, except for the top part of (**a**).

compute feature importance measures for the genotypes and knockoffs. The contrast between the importance of each variant and its knockoff is used to compute a test statistic in each LD block, for which a significance threshold is calibrated by the *knockoff filter*[36]. As a predictive model, we adopt efficient implementations of sparse linear and logistic regression designed for massive datasets[38], although other methods could be easily incorporated[18]. Thus, we can exploit the power of variable selection techniques that would otherwise offer no statistical guarantees[18,31]. This methodology is detailed in the "Methods". A schematic of our workflow is shown in Supplementary Fig. 4, while software and tutorials are available from https://msesia. github.io/knockoffzoom. The computational cost (Supplementary Note 3, Supplementary Table 2) compares favorably to that of alternatives, e.g., BOLT-LMM[7,8].

Revisiting Fig. 1, note that our discoveries are clearly interpretable as they are distinct by construction. Hypotheses at different resolutions are easily reconciled since our partitions are nested (each block is contained in exactly one block from the resolution below), while the null hypothesis for a group is true if and only if all of its subgroups are null[39]. Most findings are confirmed by those at lower resolution, even though this is not explicitly enforced, and some "floating" blocks are occasionally reported (see Fig. 1). These may be false positives or correspond to true subthreshold signals at lower resolution. A variation of *KnockoffZoom* can explicitly avoid "floating" blocks by coordinating discoveries at multiple resolutions, although with some power loss (Supplementary Note 4).

While our final output is a set of distinct discoveries that controls the FDR at each resolution, we can also quantify the statistical significance of individual findings, if these are sufficiently numerous, by estimating a local version of the FDR[40] (Supplementary Note 5).

Our findings lend themselves well to cross-referencing with gene locations and functional annotations, as shown below in our results. By contrast, the LMM output in Fig. 1c is less informative: many clumps reported by the standard PLINK[12] algorithm are difficult to interpret because they are wide and overlapping. This problem is clearer in simulations where we know the causal variants, as in Fig. 2. Here, stronger signals have two consequences: *KnockoffZoom* precisely identifies the causal variants, while the LMM reports broader and increasingly contaminated associations.

**Conditional hypotheses and population structure**. *KnockoffZoom* discoveries, by accounting for LD, bring us closer to the identification of functional variants; therefore, they are more interesting than marginal associations. Moreover, while other methods must account for population structure via principal component analysis[41] or an LMM[3–5], our conditional hypotheses are naturally less susceptible to such confounding[29] since they already account for the information in all genotypes, which includes the population structure[4,5,42]. Our hypotheses are most robust at the highest resolution, where we condition on all SNPs except one. The robustness may decrease as more variants are grouped and removed from the conditioning set. However, we

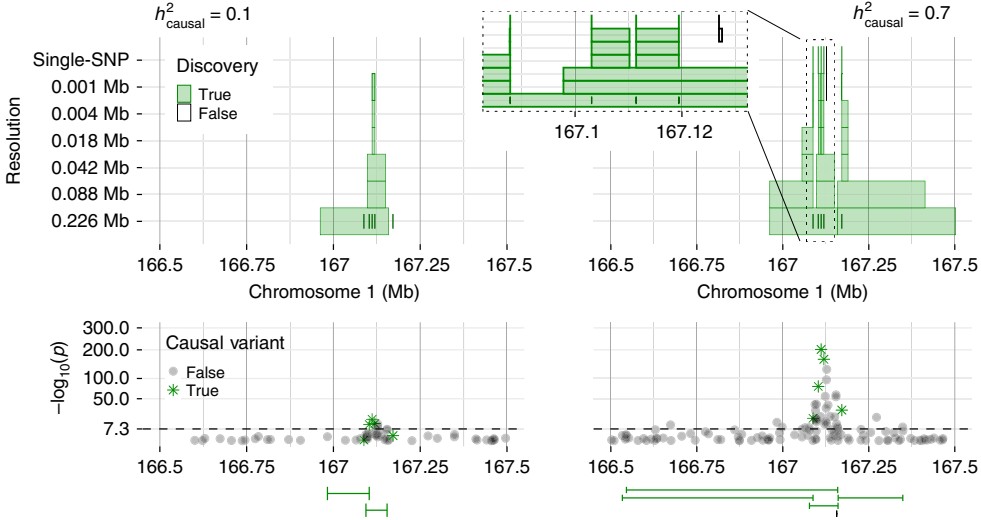

**Fig. 2 Discoveries for simulated traits.** *KnockoffZoom* discoveries for two simulated traits with the same genetic architecture but different heritability $h^2_{causal}$, within a genomic window. Other details are presented as in Fig. 1. This region contains five causal SNPs whose positions are marked by the short vertical segments at the lowest resolution. The zoomed-in view (right) shows the correct localization of the causal SNPs, as well as a "floating" false positive.

always consider fairly small LD blocks, typically containing less than 0.01% of the SNPs, even at the lowest resolution. By comparison, LMMs may not account for an entire chromosome, to avoid "proximal contamination"[7].

Our inferences rely on a model for the distribution of the genotypes, which requires some approximation. The HMM implemented here[21] is more suitable to describe homogeneous and unrelated individuals because it does not capture long-range dependencies, which is a limitation we will lift in the future. Meanwhile, we analyze unrelated British individuals, for which we verify the robustness of our approximation with simulations involving real genotypes. In the analysis of the UK Biobank phenotypes, our results already explicitly account for any available covariates, including the top principal components of the genetic matrix. Moreover, one could account for any known structure (labeled populations) with the current HMM by fitting separate models and generating knockoffs independently for each population.

**Missing and imputed variants**. In this paper, we analyze SNP data that only include a fraction of all variants. Therefore, our conditional hypotheses can localize important effects, but they cannot identify exactly the biologically causal variants, which may be missing. Only in the simulations below, where the causal variants are not missing, we can verify that *KnockoffZoom* identifies them exactly while controlling the FDR. In practice, one could impute the missing variants and then analyze them as if they were measured. However, while meaningful for marginal tests, this would not be useful to study conditional association because imputed variants contain no information about any phenotype beyond that carried by the genotyped SNPs. In fact, the imputed values are conditionally independent of any phenotype given the genotyped SNPs, since they are a function of the SNP data and of an LD model estimated from a separate panel of independent individuals. Therefore, without strict modeling assumptions (Supplementary Note 6), it is impossible to determine whether the causal variant is the missing one or among those used to impute it.

**Performance in simulations**. Setup: We compare *KnockoffZoom* with state-of-the-art methods via simulations based on 591k

SNPs from 350k unrelated British individuals in the UK Biobank (Methods). We simulate traits using a linear model with Gaussian errors: 2500 causal variants are on chromosomes 1–22, clustered in evenly spaced 0.1-Mb-wide groups of 5 (for other architectures, see Supplementary Table 3). The heritability is varied as a control parameter. This architecture is likely too simple to be realistic[11], but it facilitates the comparison with other tools by accommodating their assumptions. By contrast, we are not protecting *KnockoffZoom* against model misspecification because we use real genotypes and approximate their distribution with the same HMM as in the analysis of the UK Biobank phenotypes (our sole assumption is that we can do this accurately). Therefore, these simulations are unrealistic only with regard to the conditional distribution of the trait given the genotypes, about which *KnockoffZoom* makes no assumptions. Thus, we explicitly demonstrate our robustness to model misspecification, including the possible presence of some population structure, and the lower accuracy of the HMM for rarer variants (Supplementary Fig. 3).

In this setup, the tasks of locus discovery and fine mapping are clearly differentiated. The goal of the former is to detect broad genomic regions that contain signals; also, scientists may wish to count distinct associations. The goal of the latter is to identify the causal variants. Here, there are 500 interesting regions and 2500 signals. For locus discovery, we compare *KnockoffZoom* at low resolution to BOLT-LMM[8]. For fine mapping, we compare *KnockoffZoom* at 7 levels of resolution (Supplementary Table 1) to CAVIAR[14] and SUSIE[17] (Supplementary Note 7). For each method, we report the power, false-discovery proportion (FDP) and mean width of the discoveries (distance in base pairs between the leftmost and rightmost SNPs). Since locus discovery and fine mapping have different goals, we will need to define false positives and count the findings appropriately. Simulations with explicit coordination across different resolutions are discussed in Supplementary Fig. 5. Details regarding our code, relevant third-party software, and tuning parameters are given in Supplementary Note 8.

Locus discovery: We apply *KnockoffZoom* at low resolution, with typically 0.226-Mb-wide LD blocks, targeting an FDR of 0.1. For BOLT-LMM[8], we use the standard Bonferroni threshold $5 \times 10^{-8}$ to control the familywise error rate (FWER) below 0.05, for the marginal hypotheses.

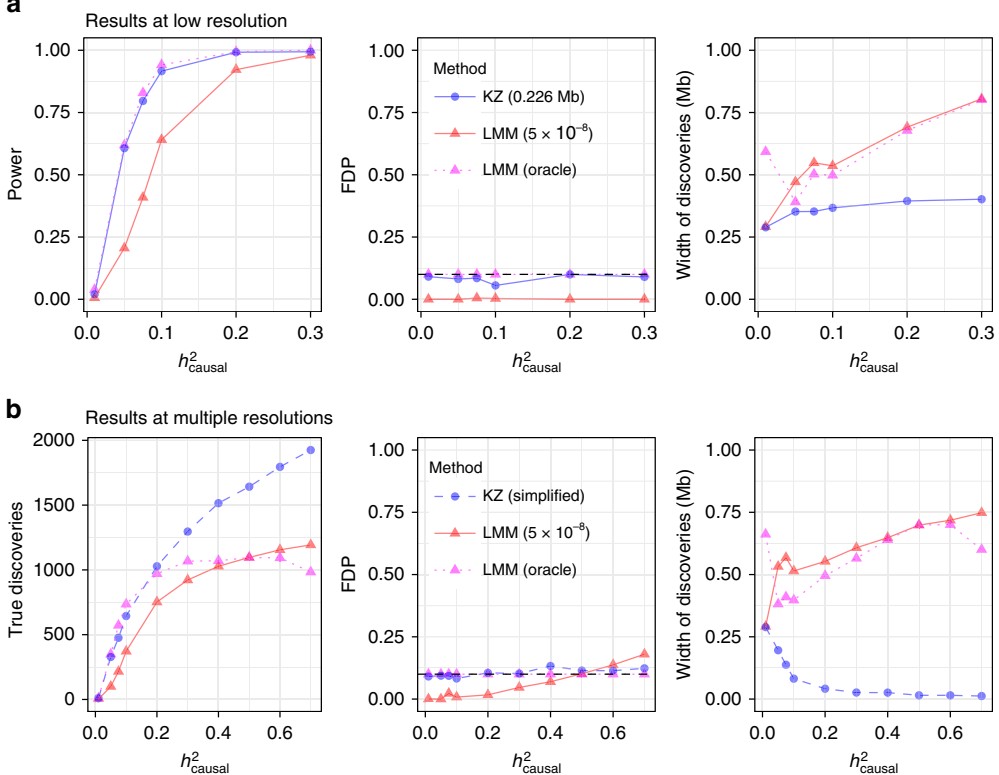

**Fig. 3 Locus discovery in simulations.** Locus discovery for a simulated trait with *KnockoffZoom* (nominal FDR 0.1) and BOLT-LMM ($5 \times 10^{-8}$ and oracle). **a** Low-resolution *KnockoffZoom* and strongly clumped LMM *p* values. **b** Multi-resolution *KnockoffZoom* (simplified count) and weakly clumped LMM.

To assess power, any of the 500 interesting regions is detected if there is at least one finding within 0.1 Mb of a causal SNP. This choice favors BOLT-LMM, which reports wider findings (Fig. 3). To evaluate the FDP, we count the false discoveries. This is easy with *KnockoffZoom* because its findings are distinct and count as false positives if causal variants are not included. In comparison, distinct LMM discoveries are more difficult to define. Following the typical approach, we clump together nearby significant variants with PLINK, using standard parameters[8] (Supplementary Note 8). Then, we define the FDP as the fraction of clumps whose range does not cover a causal SNP. For locus discovery, we consolidate clumps within 0.1 Mb.

*KnockoffZoom* and BOLT-LMM target different error rates, complicating the comparison. Ideally, we would like to control the FDR of the distinct LMM findings. Unfortunately, this is difficult[35] (Supplementary Note 9, Supplementary Fig. 6). Within simulations, we can circumvent this obstacle with a hypothetical LMM oracle that is guaranteed to control the FDR. The oracle knows which SNPs are causal, and uses this information to identify the most liberal *p* value threshold such that the FDP is below 0.1 (Supplementary Note 8). Clearly, this oracle is not practical. However, its power provides an informative upper bound for any future FDR-controlling procedure based on BOLT-LMM *p* values.

Figure 3a compares these methods as a function of the heritability. The results refer to an independent experiment for each heritability value. The average behavior across repeated experiments is discussed in Supplementary Figs. 7 and 8. The FDP of *KnockoffZoom* is below the nominal level (the FDP should be controlled on average), while its power is comparable to that of the oracle. Our method reports more precise discoveries, while the BOLT-LMM discoveries become wider as the signal strength increases, as anticipated from Fig. 2. Moreover, the standard *p* value threshold is substantially less powerful.

Before turning to fine mapping, we compare *KnockoffZoom* and BOLT-LMM at higher resolutions. Above, the aggressive LMM clumping strategy is informed by the known locations of the regions containing causal variants, to ensure that each is reported at most once. However, this information is generally unavailable, and the scientists must determine which discoveries are important. Therefore, we set out to find as many distinct associations as possible, applying *KnockoffZoom* at multiple resolutions, and interpreting the PLINK clumps as distinct findings, without consolidation. Figure 2 visualizes an example of the outcomes. Here, we follow a stricter definition of type-I errors: a discovery is true if and only if the reported set of SNPs includes a causal variant. We measure power as the number of true discoveries. Instead of showing the *KnockoffZoom* results at each resolution (Supplementary Fig. 9), we count only the most specific findings whenever the same locus is detected at multiple resolutions, and discard finer discoveries that are unsupported at lower resolutions (simplified count). This operation is unnecessary to interpret our results, and is not sustained by theoretical guarantees[43], although it is quite natural and informative.

Figure 3b shows that *KnockoffZoom* reports increasingly precise discoveries as the signals grow stronger, while the ability of BOLT-LMM to resolve distinct signals worsens. As stronger signals make more noncausal variants marginally significant through LD, the interpretation of the marginal *p* values becomes more opaque, and counting different clumps as distinct discoveries clearly leads to an excess of false positives. For this reason, the oracle procedure must become more conservative and report fewer discoveries when the signals are strong. Overall, this cautions one against placing too much confidence in the estimated number of distinct findings obtained with BOLT-LMM via clumping[8].

Fine mapping: After applying BOLT-LMM ($5 \times 10^{-8}$ significance), we separately fine-map each associated region with either

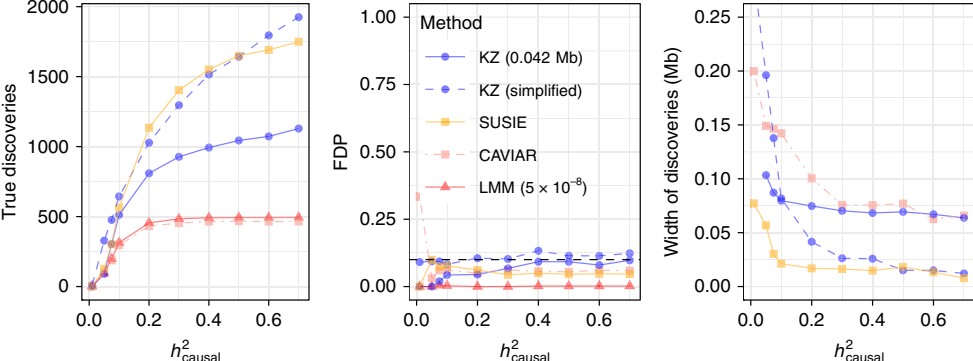

**Fig. 4 Fine mapping in simulations.** *KnockoffZoom* compared with a two-step fine-mapping procedure consisting of BOLT-LMM followed by CAVIAR or SUSIE, in the same simulations as in Fig. 3. Our method, CAVIAR, and SUSIE control a similar notion of FDR at the nominal level 0.1.

**Table 1 Discoveries at different resolutions. Numbers of distinct findings made by *KnockoffZoom* at different resolutions (FDR 0.1), on some phenotypes in the UK Biobank, compared with BOLT-LMM ($p$ values $\leq 5 \times 10^{-8}$). Data from 350 k unrelated British individuals. The LMM discoveries are counted with two clumping heuristics, as in Fig. 3.**

| Phenotype | *KnockoffZoom* | | | | | | | BOLT-LMM | |
|---|---|---|---|---|---|---|---|---|---|
| | Resolution | | | | | | | | |
| | 0.226 Mb | 0.088 Mb | 0.042 Mb | 0.018 Mb | 0.004 Mb | 0.001 Mb | Single SNP | Clumped | Clumped and consolidated |
| Height | 3284 | 1976 | 823 | 388 | 336 | 170 | 173 | 1685 | 795 |
| bmi | 1804 | 555 | 60 | 33 | 24 | 0 | 15 | 389 | 328 |
| Platelet | 1460 | 890 | 408 | 276 | 161 | 181 | 143 | 723 | 428 |
| sbp | 722 | 297 | 95 | 0 | 0 | 0 | 0 | 197 | 178 |
| cvd | 514 | 182 | 51 | 0 | 0 | 0 | 0 | 156 | 136 |
| Hypothyroidism | 212 | 108 | 0 | 0 | 0 | 0 | 21 | 96 | 77 |
| Respiratory | 176 | 65 | 41 | 13 | 14 | 12 | 0 | 63 | 47 |
| Diabetes | 50 | 33 | 21 | 10 | 11 | 10 | 0 | 47 | 42 |
| Glaucoma | 0 | 0 | 0 | 0 | 0 | 0 | 0 | 5 | 5 |

CAVIAR[14] or SUSIE[17]. We aggressively clump the LMM findings to ensure that they are distinct, as in Fig. 3a. We also provide unreported nearby SNPs as input to SUSIE, to attenuate the selection bias (Supplementary Fig. 10). Within each region, CAVIAR and SUSIE report sets of SNPs that are likely to contain causal variants. We tune their parameters to obtain a genome-wide FDR comparable to our target of 0.1 (Supplementary Note 8).

The results are shown in Fig. 4, defining the power and FDP as in Fig. 3b. The output of *KnockoffZoom* is presented in two ways: at fixed resolution (e.g., 0.042 Mb, see Supplementary Table 1) and summarizing the results by counting only the highest-resolution discoveries in each locus, as in Fig. 3b. Again, this simplified count is a useful summary, even though theoretically we can only control the FDR at each resolution separately. All methods appear to control the FDR and detect the 500 interesting regions if the heritability is sufficiently large. Moreover, they report precise discoveries, each including only a few SNPs. CAVIAR cannot make more than 500 discoveries here, as it is unable to distinguish between multiple causal variants in the same locus[17]. SUSIE and *KnockoffZoom* identify more distinct signals, with comparable performance. However, *KnockoffZoom* can be more powerful in other settings (Supplementary Fig. 11).

Note that *KnockoffZoom* tests predefined hypotheses at each resolution, while CAVIAR and SUSIE can report any set of noncontiguous SNPs. Therefore, *KnockoffZoom* may group together nearby SNPs that are not as highly correlated with each other as those grouped together by the other methods, especially

when the signals are weak (Supplementary Fig. 12). Although contiguity is not required in principle, we find it to be meaningful if not all variants are genotyped (Supplementary Note 10).

Finally, we can also show that *KnockoffZoom* is robust to smaller allele frequencies within these simulations (Supplementary Fig. 13), and can provide accurate estimates of the individual significance of each discovery (Supplementary Fig. 14).

**Analysis of different traits in the UK Biobank data.** Findings: We study four continuous traits and five diseases in the UK Biobank (Supplementary Table 4) with *KnockoffZoom*, using the same SNPs from 350k unrelated British individuals (Methods) and the same knockoffs as in the simulations. Our discoveries are compared with the BOLT-LMM findings in Table 1. We account for the principal components[41] and other covariates in the predictive model (Methods). The results remain stable if we ignore the principal components (Supplementary Table 5), confirming the intrinsic robustness to population structure. *KnockoffZoom* misses very few of the BOLT-LMM discoveries (see Methods about glaucoma) and reports many additional findings (Supplementary Table 6), even when the latter is applied to a larger sample[8] (Supplementary Table 7). The interpretation of the LMM findings is unclear because many PLINK clumps are overlapping, as shown in Fig. 1. If we consolidate them, they become distinct but less numerous, inconsistently with the results reported by others using the same data[8].

We usually obtain fewer findings at higher resolutions, since these conditional hypotheses are more challenging, but there are

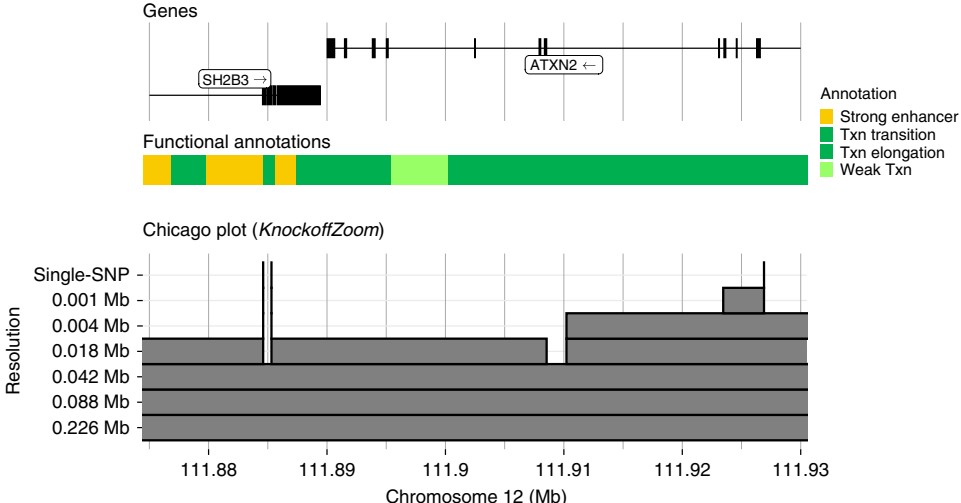

**Fig. 5 High-resolution discoveries for platelet count.** Visualization of some discoveries made with *KnockoffZoom* for platelet in the UK Biobank, along with gene positions and functional annotations (ChromHMM[52], GM12878 cell line) in the LocusZoom[53] style. The three discoveries at single-variant resolution are labeled. Other details are given as in Fig. 1.

exceptions. Some exceptions may be due to multiple nearby signals (as in Fig. 2), while others may be due to random variability in the analyses at different resolutions. The results obtained by coordinating discoveries at different resolutions are given in Supplementary Table 8. The individual significance of each discovery can be evaluated through the local FDR (Supplementary Fig. 15). Our findings can be compared with gene positions and functional annotations to shed more light onto the causal variants. We provide an online tool to explore these results interactively (https://msesia.github.io/knockoffzoom/ukbiobank); see Fig. 5, Supplementary Fig. 16 for examples. We report the allele frequencies of the variants selected by *Knockoff-Zoom* at low resolution (0.226 Mb) for different traits in Supplementary Table 9.

Reproducibility: To investigate reproducibility, we study height and platelet on a subset of 30 k individuals, and verify that the low-resolution discoveries are consistent with those previously reported for BOLT-LMM applied to all 459 k European subjects.[8] We say that a discovery is replicated if it is within 0.1 Mb of a SNP with a $p$ value below $5 \times 10^{-9}$ on the larger dataset. We do not consider the other phenotypes, for which both methods make fewer than ten discoveries with this small sample. The LMM findings are clumped by PLINK without consolidation, although this makes little difference because extensive overlap occurs only with larger samples. To illustrate the difficulty of controlling the FDR with the LMM (Supplementary Note 9), we also naively apply a pre-clumping Benjamini–Hochberg (BH) correction[19] to the LMM $p$ values.

The results are summarized in Table 2 and Supplementary Table 10. All BOLT-LMM discoveries in the smaller dataset ($5 \times 10^{-8}$) are replicated, at the cost of lower power, while the BH procedure does not control the FDR. See Supplementary Table 11 for more information about power.

As a preliminary verification of the biological validity of our findings, we conduct a gene ontology enrichment analysis of the discoveries for platelet using GREAT[44]. The enrichment among five relevant terms is highly significant (Supplementary Table 12) and strengthens at higher resolutions, suggesting increasingly precise localization of causal variants.

Finally, we cross-reference with the literature the highest-resolution *KnockoffZoom* discoveries for platelet. Three findings are shown in Fig. 5: rs3184504 (missense, *SH2B3* gene),

**Table 2 Reproducibility of the low-resolution discoveries.** Reproducibility of the low-resolution discoveries made with *KnockoffZoom* (0.226 Mb) and BOLT-LMM on height and platelet, using 30 k individuals in the UK Biobank. The target FDR is 10%.

| Phenotype | Method | Discoveries | | |
| | | # | Not replicated | Size (Mb) |
|---|---|---|---|---|
| Height | *KnockoffZoom* (FDR 0.1) | 121 | 8 (6.6%) | 0.308 |
| | LMM ($5 \times 10^{-8}$) | 54 | 0 (0.0%) | 0.965 |
| | LMM-BH (FDR 0.1) | 714 | 203 (28.4%) | 0.379 |
| Platelet count | *KnockoffZoom* (FDR 0.1) | 81 | 5 (6.2%) | 0.319 |
| | LMM ($5 \times 10^{-8}$) | 47 | 0 (0.0%) | 0.674 |
| | LMM-BH (FDR 0.1) | 272 | 92 (33.8%) | 0.433 |

rs72650673 (missense, *SH2B3* gene), and rs1029388 (intron, *ATXN2* gene). Many other discoveries are in coding regions and may plausibly localize a direct functional effect on the trait (Supplementary Table 13). Some are already known to be associated with platelet (Supplementary Table 14), while others may be new findings. In particular, six missense variants had not been previously reported to be associated with platelet (Supplementary Table 15).

## Discussion

The goal of genetic mapping is to localize the variants that influence a trait. Geneticists have widely sought this goal with a two-step strategy, partly due to computational limitations, in an attempt to control type-I errors while achieving high power. First, all variants are probed to identify promising regions, without accounting for LD. To reduce false positives, a Bonferroni correction is applied to the marginal $p$ values. Then, each associated region is separately fined-mapped. This strategy is suboptimal. Indeed, if the phenotypes are influenced by hundreds of variants, the FWER is too stringent and inhibits power unnecessarily. This error rate is a legacy of the earlier studies of Mendelian diseases, and it has been retained for complex traits mostly due to methodological difficulties, rather than a true need to avoid any false

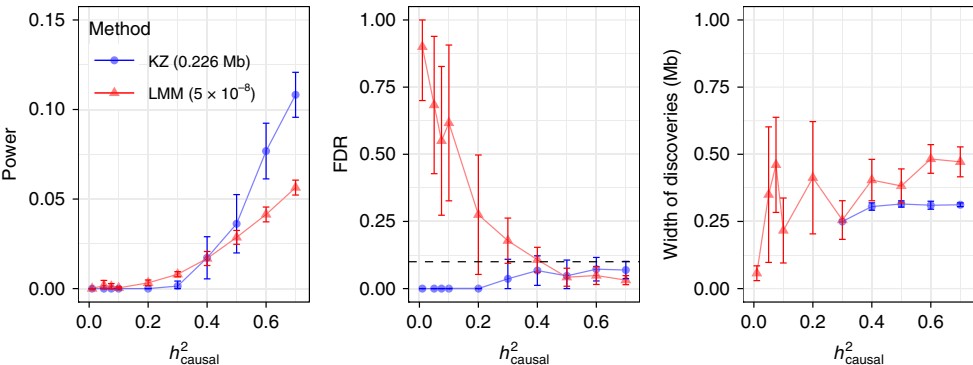

**Fig. 6 Locus discovery in a simulated case–control study.** Locus discovery with *KnockoffZoom* and BOLT-LMM for a simulated case–control study. There are 500 evenly spaced causal variants. The error bars indicate 95% confidence intervals for the mean power and FDR, as defined in Fig. 3a, averaging 10 independent replications of the trait given the same genotypes.

positives. In fact, we have shown that the two-step paradigm of locus discovery followed by fine mapping already effectively tries to control an error rate comparable to the FDR that we directly target. Besides, the FWER guarantee is only valid for the SNP-by-SNP marginal hypotheses of no association. These are of little interest to practitioners, who typically interpret the findings as if they contained causal variants. By contrast, *KnockoffZoom* unifies locus discovery and fine mapping into a coherent statistical framework, so that the findings are immediately interpretable and equipped with solid guarantees.

We are not the first to propose a multi-marker approach to genetic mapping[31,35,42,45–47], or to consider testing groups of SNPs at different resolutions[48], although knockoffs finally allow us to provide solid type-I error guarantees based on relatively realistic assumptions.

For ease of comparison, we have simulated phenotypes using a linear model with Gaussian errors, which satisfies many of the standard assumptions. Unfortunately, there is little information on how real traits depend on the genetic variants; after all, the goal of a GWAS is to discover this. Therefore, relying on these assumptions can be misleading. In contrast, *KnockoffZoom* only relies on knowledge of the genotype distribution, which we can estimate accurately due to the availability of genotypes from millions of individuals. Indeed, geneticists have developed phenomenological HMMs of LD that work well for phasing and imputation.

We highlight with an example that our framework is not tied to any model for the dependence of phenotypes on genotypes, or to any specific data analysis tool. Therefore, we simulate an imbalanced case–control study where BOLT-LMM is known to fail[49], while our method applies seamlessly. This trait is generated from a liability threshold model (probit), obtaining 525 cases and 349,594 controls. We apply *KnockoffZoom* using sparse logistic regression, and report the results in Fig. 6 as a function of the heritability of the latent Gaussian variable.

Like many other modern statistical methods, *KnockoffZoom* is randomized: its output depends on random variables that are not part of the data, in this case the knockoffs. Even though we have not done it here, it is possible to re-sample the knockoffs and obtain different sets of discoveries, each controlling the FDR[18,24]. However, it is not yet clear how to best combine these while preserving the FDR control. This is challenging in theory and it should be studied further. Meanwhile, we have repeated our analyses using a different set of knockoffs in Supplementary Table 16, and verified that the discoveries are relatively stable, especially when their number is large. It is worth mentioning that the stabilities of our discoveries can be predicted well from their individual statistical significance, which we can estimate with a

local version of the FDR (Supplementary Tables 17–20). Supplementary Table 21 shows that most of the significant loci identified by BOLT-LMM are consistently reported by *KnockoffZoom*. Instead, the variability in *KnockoffZoom* typically involves either completely new loci (near the FDR threshold) or finer-grained details (multiple distinct discoveries within the same locus) that cannot be detected by BOLT-LMM.

Our software (https://msesia.github.io/knockoffzoom) has a modular structure that accommodates many options, reflecting the flexibility of *KnockoffZoom*. Users may experiment with different importance measures: for example, one can incorporate prior information, such as summary statistics from other studies. Similarly, there are many ways of defining the LD blocks: leveraging genomic annotations is a promising direction.

We are working on more refined knockoff constructions, with new algorithms for heterogeneous populations and rarer variants based on an HMM similar to that of SHAPEIT[23]. Similarly, we are developing new algorithms for family data. In the future, we plan to analyze even more variants, possibly from sequencing studies. This will involve additional computational challenges, but it is feasible in principle since our algorithms are parallelizable and scale linearly with the data size.

Finally, *KnockoffZoom* may lead to more principled polygenic risk scores. Partly with this goal, multi-marker analyses of GWAS data have been suggested long before our contribution[31,42,45–47,50,51]. However, knockoffs finally allow us to obtain reproducible findings with type-I error guarantees.

## Methods

**Formally defining the objective of *KnockoffZoom*.** We observe a phenotype $Y \in \mathbb{R}$ and genotypes $\mathbf{X} = (X_1, \dots, X_p) \in \{0, 1, 2\}^p$ for each individual. We assume that the pairs $(\mathbf{X}^{(i)}, Y^{(i)})_{i=1}^n$ corresponding to $n$ subjects are independently sampled from some distribution $P_{\mathbf{X}Y}$. The goal is to infer how $P_{Y|\mathbf{X}}$ depends on $\mathbf{X}$, testing the conditional hypotheses defined below, without assuming anything else about this likelihood, or restricting the sizes of $n$ and $p$. Later, we will describe how we can achieve this by leveraging prior knowledge of the genotype distribution $P_{\mathbf{X}}$. Now, we formally define the hypotheses.

Let $\mathcal{G} = (G_1, \dots, G_L)$ be a partition of $\{1, \dots, p\}$ into $L$ blocks, for some $L \le p$. For any $g \le L$, we say that the $g$th group of variables $\mathbf{X}_{G_g} = \{X_j | j \in G_g\}$ is null if $Y$ is independent of $\mathbf{X}_{G_g}$ given $\mathbf{X}_{-G_g}$ ($\mathbf{X}_{-G_g}$ contains all variables except those in $G_g$). We denote by $\mathcal{H}_0 \subseteq \{1, \dots, L\}$ the subset of null hypotheses that are true. Conversely, groups containing causal variants do not belong to $\mathcal{H}_0$. For example, if $P_{Y|\mathbf{X}}$ is a linear model, $\mathcal{H}_0$ collects the groups in $\mathcal{G}$ whose true coefficients are all zero (if there are no perfectly correlated variables in different groups[39]). In general, we want to select a subset $\hat{S} \subseteq \{1, \dots, L\}$ as large as possible, and such that

$$\text{FDR} = \mathbb{E}\left[|\hat{S} \cap \mathcal{H}_0| / \max(1, |\hat{S}|)\right] \le q,$$ for a fixed partition $\mathcal{G}$ of the variants. These conditional hypotheses generalize those defined earlier in the statistical literature[18,24], which only considered the variables one by one. Our hypotheses are better suited for the analysis of GWAS data because they allow us to deal with LD without pruning the variants (Supplementary Methods). As a comparison, the null

statement of the typical hypothesis in a GWAS is that $Y$ is marginally independent of $X_j$, for a given SNP $j$.

**The knockoffs methodology**. *Knockoffs*[18] solve the problem defined above if $P_\mathbf{X}$ is known and tractable, as explained below. The idea is to augment the data with synthetic variables, one for each genetic variant. We know that the knockoffs are null because we create them without looking at $Y$. Moreover, we construct them so that they behave similarly to the SNPs in null groups, and can serve as negative controls. The original work considered explicitly only the case of a trivial partition $\mathcal{G}$ into $p$ singletons[18], but we extend it for our problem by leveraging some previous work along this direction[37,39]. Formally, we say that $\tilde{\mathbf{X}} = (\tilde{X}_1, \dots, \tilde{X}_p)$ is a group knockoff of $\mathbf{X}$ for a partition $\mathcal{G}$ of $\{1, \dots, p\}$ if two conditions are satisfied: (1) $Y$ is independent of $\tilde{\mathbf{X}}$ given $\mathbf{X}$; (2) the joint distribution of $(\mathbf{X}, \tilde{\mathbf{X}})$ is unchanged when $\{X_j : j \in G\}$ is swapped with $\{\tilde{X}_j : j \in G\}$, for any group $G \in \mathcal{G}$. The second condition is generally difficult to satisfy (unless $\tilde{\mathbf{X}} = \mathbf{X}$, which yields no power), depending on the form of $P_\mathbf{X}$[18]. In the Supplementary Methods, we develop algorithms to generate powerful group knockoffs when $P_\mathbf{X}$ is an HMM, the parameters of which are fitted on the available data using fastPHASE[21]; see Supplementary Note 1, Supplementary Figs. 2 and 3, and Supplementary Table 22 for more details about the model estimation and its goodness of fit. Here, we take $\tilde{\mathbf{X}}$ as given and discuss how to test the conditional hypotheses. For the $g$th group in $\mathcal{G}$, we compute feature importance measures $T_g$ and $\tilde{T}_g$ for $\{X_j : j \in G_g\}$ and $\{\tilde{X}_j : j \in G_g\}$, respectively. Concretely, we fit a sparse linear (or logistic) regression model[38] for $Y$ given $[\mathbf{X}, \tilde{\mathbf{X}}] \in \mathbb{R}^{n \times 2p}$, standardizing $\mathbf{X}$ and $\tilde{\mathbf{X}}$; then we define $T_g = \sum_{j \in G_g} |\hat{\beta}_j(\lambda_{\text{CV}})|$, $\tilde{T}_g = \sum_{j \in G_g} |\hat{\beta}_{j+p}(\lambda_{\text{CV}})|$. Above, $\hat{\beta}_j(\lambda_{\text{CV}})$ and $\hat{\beta}_{j+p}(\lambda_{\text{CV}})$ indicate the estimated coefficients for $X_j$ and $\tilde{X}_j$, respectively, with regularization parameter $\lambda_{\text{CV}}$ tuned by cross-validation. These statistics are designed to detect sparse signals in a generalized linear model—a popular approximation of the distribution of $Y$ in a GWAS[31]. Our power may be affected if this model is misspecified, but our inferences remain valid. A variety of other tools could be used to compute more flexible or powerful statistics, perhaps by incorporating prior knowledge[18]. Finally, we combine the importance measures into test statistics $W_g$ through an antisymmetric function, e.g., $W_g = T_g - \tilde{T}_g$, and report groups of SNPs with sufficiently large statistics[18]. The appropriate threshold for FDR control is calculated by the knockoff filter[36]. Further details about the test statistics are given in Supplementary Note 11.

As currently implemented, our procedure has no power at the nominal FDR level $q$ if there are fewer than $1/q$ findings to be made. Usually, this is not a problem for the analysis of complex traits, where many loci are significant. However, this may explain why, at level $q = 0.1$, we report none of the 5 discoveries obtained by BOLT-LMM for glaucoma in Table 1. Alternatively, we may detect these by slightly relaxing the knockoff filter[36], at the cost of losing the provable FDR guarantee.

**Including additional covariates**. We control for the sex, age, and squared age of the subjects to increase power (squared age is not used for height, as in earlier work[8]). We leverage these covariates by including them in the predictive model for the *KnockoffZoom* test statistics, along with the top five principal components of the genetic matrix. We fit a sparse regression model on the augmented matrix of explanatory variables $[\mathbf{Z}, \mathbf{X}, \tilde{\mathbf{X}}] \in \mathbb{R}^{n \times (m+2p)}$, where $\mathbf{Z}, \mathbf{X}, \tilde{\mathbf{X}}$ contain the $m$ covariates, the genotypes, and their knockoff copies, respectively. The coefficients for $\mathbf{Z}$ are not regularized, and we ignore them in the final computation of the test statistics.

**Quality control and data preprocessing for the UK Biobank**. We consider 430,287 genotyped and phased subjects with British ancestry. According to the UK Biobank, 147,718 of these have at least one close relative in the dataset; we keep one from each of the 60,169 familial groups, chosen to minimize the missing phenotypes. This yields 350,119 unrelated subjects. We only analyze biallelic SNPs with minor allele frequency above 0.1% and in Hardy–Weinberg equilibrium ($10^{-6}$), among our 350,119 individuals. The final SNP count is 591,513. A few subjects withdrew consent and we removed their observations from the analysis.

**Additional details regarding the simulations**. The effect sizes of the causal variants are heterogeneous, with relative values chosen uniformly at random across clusters so that the ratio between the smallest and the largest is 1/19. The causal variables in the model are standardized, so rarer variants have stronger effects.

## Data availability
Data are available from the UK Biobank Resource (application 27837), see https://www.ukbiobank.ac.uk/.

## Code availability
The *KnockoffZoom* code is available from https://github.com/msesia/knockoffzoom [10.5281/zenodo.3625250]. The code for the simulations and analysis is available at https://github.com/msesia/ukbiobank_knockoffs [10.5281/zenodo.3625252].

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

## Acknowledgements
M.S., S.B., E.C., and C.S. are supported by NSF grant DMS 1712800. E.C. and C.S. are also supported by NSF grant 1934578 and by a Math+X grant (Simons Foundation). E.K. is supported by the Hertz Foundation. S.B. is supported by a Ric Weiland fellowship. Finally, we thank Sahitya Mantravadi, Ananthakrishnan Ganesan, Robert Tibshirani, and Trevor Hastie.

## Author contributions
M.S. developed the methodology, designed the experiments, and performed the data analysis. E.K. contributed to some aspects of the analysis (quality control, phenotype extraction, and SNP annotations), and the development of the visualization tool. S.B. contributed to the data acquisition. E.C. and C.S. supervised this project. All authors contributed to writing this paper.

## Competing interests
The authors declare no competing interests.
