## [Peer Review File · Nature Communications]

Reviewers' Comments:

Reviewer #1:

Remarks to the Author:

Review of Sesia et al

Summary

This interesting and impressive paper presents extensions, implementation and application of a recently-developed statistical methodology (the knockoff filter) to large GWAS (UK Biobank). The methods provide guaranteed control of False Discovery Rates when testing pre-specified contiguous groups of SNPs (or other variants). Importantly, the null hypothesis being tested here is not the commonly-used null that the group of SNPs is *marginally* unassociated with the trait; instead the null is that the group is *conditionally* unassociated with the trait given all other observed SNPs. This conditional test is in many ways more informative than conventional marginal tests because it ensures that a significant group cannot be explained by linkage disequilibrium (LD) with other measured SNPs outside the group. Thus the conditional test comes closer to identifying groups of potentially-causal SNPs than do conventional marginal tests.

The paper is very well presented, and the results and comparisons with other methods seem generally appropriate and interesting. My main request is that the paper should better highlight the limitations of the method -- specifically, at high resolution ("fine-mapping") the need to confine tests to pre-specified contiguous groups of SNPs seems a clear disadvantage compared with existing fine-mapping methods. This is not to take away from the other important contributions of this work.

Major Comment

As mentioned above, the main limitation of the current implementation (and perhaps the whole framework?) is the requirement that groups of tested markers be both contiguous and pre-specified. At coarser resolutions, where the main goal is to identify genomic regions (conditionally) associated with the trait, these requirements are not a major limitation. However at fine-scale resolutions, where one is trying to get down to the likely causal markers, these requirements becomes more bothersome. For example suppose we have 4 SNPs, in order, A-B-C-D, and A and D are in very strong LD with each other (say LD of 1 for concreteness), but not in strong LD with B or C, and A is the causal SNP. Then the contiguity requirement of knockoffZoom will not allow it to refine the association beyond the entire group (A-D), even though in principle one could narrow it down further to SNPs A and D. Existing fine-mapping methods do not have this limitation and could report (A,D) as the set of potential causal markers. Further, even if the contiguity requirement were relaxed (e.g. to allow prespecified non-contiguous groups), the need to prespecify groups to be tested may still limit the resolution to which associations can be refined.

For this reason I think it is premature to claim "...KnockoffZoom unifies locus discovery and fine-mapping into a coherent statistical framework" (p15). Specifically, I think its abilities to solve the fine-mapping problem are not yet adequate to make this claim, and that studies interested in fine mapping will continue to want to use existing Bayesian fine-mapping methods like SUSIE (quite possibly as a complement to knockoffZoom) to refine associations as far as possible. In any case, the limited resolution that comes with testing contiguous pre-specified marker groups should be better highlighted in the text.

Besides better highlighting this limitation in text, the comparisons with fine-mapping methods should be extended to quantify the effect. Currently the comparisons show the "width" of region identified by each method (Figure 4, right panel). However, fine-mapping methods do not strictly identify a region but a set of SNPs, so the figure should also compare the number of SNPs identified by each method. It would also be informative to show the minimum pairwise LD between the markers identified -- does knockoffZoom sometimes report markers not in high LD with one another due to the contiguity constraint? (Incidentally, the y axis on this figure is too large to see the interesting region, which for fine-mapping is <0.1 Mb. Getting to a region of 0.5 Mb is not really fine mapping in my opinion.)

It would also be interesting to get the authors' perspective on how easy or difficult it might be for the contiguity requirement to be relaxed in the future. (Also the pre-specification requirement, although this seems more fundamental.)

Other main comments

- Some aspects of Table 1 are surprising to me. Eg the number of bmi findings going from 24 -> 0 -> 15 as resolution increases. Shouldn't power increase as larger groups are tested? (I realize there are fewer tests as groups get bigger...so this is not a simple issue.) The hypothyroidism results are perhaps even weirder. Can you provide any intuitive explanation for why this might occur? Is it simply chance, since the knockoff procedure can produce different results if run multiple times?

- The introduction criticizes the two-step approach as "not fully satisfactory because it requires switching models and assumptions in the middle of the analysis, obfuscating the interpretation of the findings and possibly invalidating type-I error guarantees." However, from Table 1 (see above comment), performing separate analyses at different resolutions appears to have similar problems regarding interpretation. The method that avoids "floating" discoveries at high resolution (Supplement S1B) seems to address this, but at a cost in power. What is that cost in power for the analyses here? How does Table 1 look if you apply that method? (with or without the 1.93 factor mentioned in the supplement).

- As I understand it the output at each resolution depends on a single

generation of the knockoff variables, and so the method will report different significant results each time it is run? Is this correct? If so, how different/similar are the results if you run things a second time with another knockoff realization? (It could suffice to do one trait twice to illustrate this)

- The notation $(X, X_{\tilde{}})$ suggests that the knockoffs are always included after the real variables in the input file to the lasso/bigsnpr. In principle the location of the knockoffs in the input file should not matter when a convex method like lasso is being applied (with the exception of variables with $LD=1$, which is already dealt with here as a special case). However, if one were to replace the lasso with non-convex methods the non-random order of the markers into the method could lead to failure to control FDR (eg if the method has a bias towards choosing columns earlier in the list of covariates). Further, even for convex methods, there is some concern numerical issues could arise to create this bias. As a safety check I suggest running the method with randomly ordered columns, or if that is too much of a pain simply reversing $(X_{\tilde{}}, X)$ to check it makes no difference.

- I found the references to the Li-Stephens model vs fastPHASE model in the Supplement confusing. The description of Li-Stephens as "This HMM describes the distribution of genotypes as a patchwork of latent ancestral motifs" is incorrect - this describes the fastPHASE model. The Li-Stephens model describes each haplotype as a patchwork of other observed haplotypes, not latent motifs. As I understand the text all the models here are essentially fastPHASE models not Li-Stephens models. Please clarify.

- The results in the supplement that reduce forward-backward calculations to $O(K)$ and $O(K^2)$ look similar to results that are already well established (e.g. Fearnhead and Donnelly, 2001, Estimating recombination rates from population genetic data, Genetics). Is there anything new here?

- Please provide more details about the comparisons with other methods, including versions of software and the settings used. Ideally the code used to run the comparisons with other methods should be made available - even without documentation this can be invaluable for others to see what was done.

Other comments/questions:

- Getting the method working on problems of UK biobank scale is impressive, even though limited to "only" 591k SNPs. Would applying to ~50 million SNPs be feasible, and require about 100 times the computation? For coarse resolution it might not matter much to include the extra SNPs, but for fine-mapping it ultimately seems important to include as many SNPs as possible.

- The paper discards tests where the knockoffs are very highly

correlated with the original variables (which makes sense as they have no power). For intuition I would be interested to see the distribution of the correlation of knockoffs with the original variables (say at the finest resolution).

- What is the MAF distribution of the variants analyzed here?
Does the method work equally well for common vs rare variants?
(I ask because the LD models may tend to work best for common variants.)

- It would perhaps be helpful to cite (and contrast with) previous work that attempts to control error rates of conditional tests of groups of variables (eg work on hierarchical testing by Yekutieli, Meinhausen, Bühlmann etc).

Minor:

p6: "by likelihood of the trait" -> "in distribution of the trait"

p11: "its intrinsic limitations discussed above" - I do not see where they were discussed.

p12: "As the resolution increases, we report fewer findings" - not always!

Table 1: I suggest giving resolution in terms of kb instead of Mb. Is 0.000 down to single SNP resolution?

p16: "possible *to* construct"

refs: markov -> Markov ; uk -> UK

This review is signed: Matthew Stephens

Reviewer #2:

Remarks to the Author:

This paper describes a new method for jointly identifying and fine-mapping causal variants in genetic association studies. This is a central problem to the field and the very novel approach taken by the authors opens a new avenue for exploration in comparison to the rather similar approaches of existing methods. Therefore, while some of the empirical benefits are modest, the paper is an important addition and will hopefully be expanded in future work. There are several additions and clarifications that would be beneficial:

The fine mapping methods compared to can explore causal variants that are not contained in contiguous groups. For example, caviar explores arbitrary assignments of causal variants across the entire associated locus. The authors should explain this component of the difference in the main text.

Along those lines: How is width defined for the results of caviar and other methods? Is it the case the width is smaller for these methods and by how much? Can you also compare on their favored of credible set? That is, can you specify how many additional variants are needed for each of the methods before the causals are included as they do in their papers? Finally, is it possible to generate knockoffs from non-contiguous groups in order to achieve better resolution?

With 350k individuals is it possible to run the desired penalized multivariate regression in the associated locus in many instances? If so, how does this approach perform relative to the others?

It is well known that the HMM models of genetic data, including Li & Stephens, are worse for rare variants. Do you see a relative decay in performance at rare variants compared to other methods?

Can you and have you explicitly tested the knockoffs generated for the properties required? In particular, have you looked at exchangeability of the rare variants?

Can the method be adapted to handle imputed data and what happens when the causal variants are in the imputed set? An examination and discussion of imputed data is warranted given its pervasiveness in GWAS.

It is worth mentioning section "Assessing the individual significance of each discovery" in the discussion because it is likely to be used in practice.

It seems likely that removing null regions from a PRS will improve prediction accuracy in a similar manner improving tests of enrichment for causal loci.

Minor Comments:

No (a) (b) (c) etc in Fig 1.

In Fig 3 (or supp) it would be helpful to see the number of false discoveries as a function of h^2 similar to the subfigure in the lower left. Unless these are identical at 10% for all methods?

I am not able to review section "Coordinating discoveries across resolutions" because I lack the background in that area of statistics.

Reviewer #3:

Remarks to the Author:

The work aims at identifying causal variants using a new analytical technique named as KnockoffZoom and "unifies locus discovery and re-mapping into a coherent statistical framework". The work is carefully presented and accompanied by a released software package, but it can be improved in several aspects.

1. I find the abstract and the introduction overstate the ability of the proposed in terms of identifying the causal variants. The latter statement that "The KnockoffZoom discoveries, by accounting for LD, bring us closer to the identification of functional variants" is much more accurate than "KnockoffZoom localizes causal variants precisely" in the abstract.

2. The disposition of the literature review in introduction can be expanded in that methods to be compared with (e.g. CAVIAR and SUSIE) are discussed explicitly.

3. The lipid illustrative example is misleading in that it implies that the traditional GWAS does not account for population structure inferred from all genotypes; incorporating PCA is a standard and effective practice that deals with the confounding issue illustrated by the lipid example.

Further, the proposed "inferences rely on a model for the distribution of the genotypes, which requires some approximation. The HMM implemented here is more suitable to describe homogeneous and unrelated individuals", while "results already explicitly account for any available covariates, including those that reflect the population structure, such as the top principal components of the genetic matrix". Thus, how the proposed "conditional hypotheses are less susceptible to the confounding effect of population structure" is not clear. The sensitivity of the

method to the HMM approximation is also not clear.

4. Figure 1 (a)-(c) annotation is missing from the plot.

5. There are some features in Figure 3 require explanations, e.g. why the width of the oracle procedure is not monotone as h_2 increases? Why the true discoveries of the oracle procedure are substantially less than the proposed method as h_2 increases in (b)? And why the FDP of LMM increases as h_2 increases in (b)?

6. Results in Figure 4 suggest that SUSIE is a better method than the proposed one with lower FDP and narrower width of discoveries. I suggest highlighting their differences in the main text instead of the Supplementary materials.

7. The analysis presented in Section D should include SUSIE in parallel with the simulation studies in Section C.

8. Figure 5 should include results from the standard analyses in parallel with the earlier figures.

9. Finally, the authors discussed extensively their choice of controlling FDR instead of FWER in the discussion section. Still, I find the choice a bit odd given the nature of the study: identifying causal variants as opposed to exploratory analyses. This difference also complicates the later comparison between KnockoffZoom and BLOT-LMM. Indeed, despite the effort of presenting an oracle procedure the results presented in Figure 3 can be misleading given the differential FDP. I do not request fundamentally changing the error control procedure, but I appreciate more insightful comments on this issue.

Multi-resolution localization of causal variants across the genome
Reply to the comments of the reviewers

Matteo Sesia, Eugene Katsevich, Stephen Bates, Emmanuel Candès, Chiara Sabatti
Stanford University, Department of Statistics, Stanford, CA 94305, USA

CONTENTS

Overview of Main Changes	4
R1. Reviewer 1	6
A. Overview	6
B. Major comment	6
1. Hypothesis pre-specification and contiguity	6
2. Fine-mapping resolution	9
C. Other main comments	11
1. Consistency of discoveries at different resolutions	11
2. Variability upon knockoff resampling	13
3. Numerical stability of test statistics	14
4. References to the implemented HMM	14
5. Analytical calculations	14
6. Additional software information	15
D. Other comments/questions	15
1. From array SNPs to full-genome data	15
2. Distribution of correlations between knockoffs and genotypes	15
3. Distribution of minor allele frequencies	16
4. Additional references	16
E. Minor comments	17
R2. Reviewer 2	18
A. Overview	18
B. Comments	19
1. Contiguous groups	19
2. Measuring resolution	19
3. Knockoffs for non-contiguous groups	20
4. Alternative two-step method	20
5. Knockoff diagnostics and rare variants	21
6. Imputed variants	22
7. Assessing the individual significance of each discovery	22
8. Towards prediction	22

C. Minor Comments	23
R3. Reviewer 3	24
A. Overview	24
B. Comments	24
1. Abstract and introduction	24
2. Population structure	25
3. Missing annotations	26
4. Additional explanations for the results of the simulations	26
5. Performance comparison with BOLT-LMM + SUSIE	26
6. Applying fine-mapping tools in data analysis	27
7. Controlling the FDR	27
References	29

OVERVIEW OF MAIN CHANGES

We thank the reviewers for the time they clearly devoted to our manuscript, and for their thoughtful comments and suggestions. We have taken their feedback to heart while carefully revising the manuscript and supplement accordingly. To facilitate the review, in what follows, we echo their original comments before our point-by-point responses. Unfortunately, this makes for a rather long reply, so we start by summarizing the main themes below.

a. Contiguous (pre-defined) SNP groups, imputed variants, and other fine-mapping methods.

By modifying both the paper and the supplement, we have clarified the conceptual and practical differences between our approach and fine-mapping methods (see Section II A, and Supplement, Section S4 G); we have explained how the choice of contiguous SNP groups is not a requirement of the knockoff approach, but results in interpretable hypotheses in presence of untyped variants (Section II C, and Supplement, Section S6 A); we have underscored how the groups of SNPs can be defined on the basis of the observed patterns of linkage disequilibrium in the genotyped data, while the multi-resolution analysis allows some level of adaptivity to the signal in the phenotype.

b. Interpretation of the multi-resolution analysis and consistency of findings across layers.

We have clarified that the discoveries at each level of resolution are considered the final output of our method. We have also explored further approaches to interpret results across different resolutions. In Section II A of the manuscript we discuss a variation of our procedure that ensures that the discoveries in all layers are consistent; that is, any discovery must be a subset of a discovery at the next-lowest resolution. In our revised supplement, we include the numerical performance of this method in simulations (Section S4 J) and on real data (Section S5 D). In the simulations, we find that the FDR is controlled, but that the power decreases. The reduction in power is very mild when using a liberal version of the procedure, and more severe when using a more conservative correction. We conclude that the “floating” discoveries can be easily avoided if desired, although they may be informative in general.

c. Knockoff construction diagnostics. In the new Section S4 C of the supplementary material (referenced in Section II D 1 of the paper), we examine the quality of the knockoffs we constructed for the UK Biobank data set. In particular, we confirm that the correlations between knockoffs and true variables match the correlations found in the original data, although we can see some limitations of the currently implemented HMM to describe long-range LD and the distribution of lower-frequency variants.

d. Impact of allele frequency. In the new Section S4 H and Section S5 G of the supplementary material, we find that *KnockoffZoom* performs well with a wide range of allele frequencies, even though we know the HMM is more accurate for more common variants. This suggests that the method is quite robust, although there is some room for further improvements, which we are addressing in current work.

e. Randomness of knockoffs and variability in discoveries To both illustrate the randomness of our procedure and evaluate its impact on discoveries, in the new Section S5 J of the supplementary material (referenced in Section III of the paper), we examine the result of *KnockoffZoom* with a second, independent set of knockoffs. We find that the discoveries overlap substantially, especially at the lowest levels of resolution. Furthermore, we show that our estimate of the local false discovery rate is a good predictor of the stability of different *KnockoffZoom* discoveries.

f. Missing and imputed variants. We have added a new section in the revised manuscript (Section II C) discussing our choice of analyzing only genotyped SNPs, as opposed to including also imputed variants. In particular, we argue that imputed variants do not contain any additional information given the genotyped variants and hence cannot increase the resolution of *KnockoffZoom*, even though they might be used to increase power. Section S6 A in the supplement also contains an expanded discussion of the role of imputed variants.

R1. REVIEWER 1

A. Overview

This interesting and impressive paper presents ... The paper is very well presented, and the results and comparisons with other methods seem generally appropriate and interesting. My main request is that the paper should better highlight the limitations ... of the method – specifically, at high resolution (“fine-mapping”) the need to confine tests to pre-specified contiguous groups of SNPs seems a clear disadvantage compared with existing fine-mapping methods. This is not to take away from the other important contributions of this work.

We are flattered by Professor Stephens’ feedback and we sincerely appreciate the suggestions. We have taken them into full account in the revised manuscript, as discussed point-by-point below.

B. Major comment

The major comment concerns the pre-specification (and contiguity) of our hypotheses. We thank Professor Stephens for bringing up this topic, since testing pre-specified hypotheses is indeed an important characteristic of our method that distinguishes it from many existing alternatives.

1. Hypothesis pre-specification and contiguity

As mentioned above, the main limitation of the current implementation (and perhaps the whole framework?) is the requirement that groups of tested markers be both contiguous and pre-specified. At coarser resolutions, where the main goal is to identify genomic regions (conditionally) associated with the trait, these requirements are not a major limitation.

We comment on the issue of pre-specification first and contiguity later.

Our group hypotheses are pre-specified in the sense that they are formulated before looking at the phenotype measurements, which is necessary for valid inference within our framework. However, this does not mean that the hypotheses should be specified without reference to the data. In fact, the observed genotypes play an important role in defining our group hypotheses. The specific choices in this paper are informed by the observed patterns of LD and the physical position of each marker, in order to obtain good power and easily interpretable results. However, *KnockoffZoom* is sufficiently flexible to allow geneticists to group the hypotheses as they find most interesting, while possibly leveraging other sources of information (e.g., recombination maps, gene

positions, variant annotations, ...). We have clarified this point in the revised Section II A.

Although the groups of SNPs in each resolution are fixed, *KnockoffZoom* also adapts to the phenotype by analyzing multiple resolutions. As we try to localize each signal at highest possible resolution, our reliance on pre-specified groups is not as limiting as it may seem at first sight. It would of course be desirable to make our procedure even more adaptive, but it is unclear whether this may be possible while rigorously controlling the FDR.

Regarding the contiguity of the groups tested by *KnockoffZoom*, this is not a requirement. *KnockoffZoom* can already be seamlessly applied to test hypotheses defined over any pre-specified genotype partitions, as we have also clarified in the revised Section II A. We simply chose to apply *KnockoffZoom* with contiguous groups in this paper for two practical reasons, as discussed next.

Firstly, we find that spatially contiguous blocks lead to the most clearly interpretable and easily visualizable results. Since genetic variants have an intrinsic sequential order and are inherited in contiguous blocks, it is intuitive to try to preserve this structure in our group hypotheses. Moreover, the patterns of LD at coarser resolutions are also roughly organized into discrete blocks,^{1,2} an observation leveraged in association tests based on haplotype blocks.³⁻⁸ Most importantly, the results of association studies based on variant-by-variant marginal testing are already typically interpreted as pointing to genetic locations rather than individual variants. The *KnockoffZoom* discoveries presented in this paper have a similar, although more precise, interpretation (see the next point for a more detailed discussion), as we have now clarified in the revised Section IID 3, and in Section S4 G 3 of the supplement.

Secondly, contiguous groups marry well with the HMM distribution we have assumed for the genotypes. While it is possible to use the same model to construct knockoffs for groups of non-contiguous SNPs, this would effectively require splitting the non-contiguous groups (Supplement, Section S2 A 1), which would result in more stringent exchangeability constraints than necessary, and thus quite possibly not lead to an increase in power, as now explained in Section II A. This is because the HMM has a well-defined spatial structure, reflecting the physical ordering of the variants along each chromosome; therefore, our knockoff generation algorithm would intrinsically treat two disjoint parts of the same non-contiguous group as if they were two different groups, whose conditional importance had to be tested separately. We cannot predict whether the power cost of this additional exchangeability would outweigh the possible benefits of having more flexible hypotheses; in practice, this may depend on the particular choice of test statistics and on the exact genetic architecture of the trait. In any case, the difficult interpretation of hypotheses defined over non-contiguous groups remains the main issue even at high resolution, as further discussed next.

However at fine-scale resolutions, where one is trying to get down to the likely causal markers, these requirements becomes more bothersome. For example suppose we have 4 SNPs, in order, A-B-C-D, and A and D are in very strong LD with each other (say LD of 1 for concreteness), but not in strong LD with B or C, and A is the causal SNP. Then the contiguity requirement of knockoffZoom will not allow it to refine the association beyond the entire group (A-D), even though in principle one could narrow it down further to SNPs A and D. Existing fine-mapping methods do not have this limitation and could report (A,D) as the set of potential causal markers.

The sequential linkage of markers on the same chromosome is not reflected exactly in the fine-scale details of the LD patterns. In fact, the various existing measures of LD (e.g., the pairwise r^2) are sensitive to allele frequencies^{9–11} and other population phenomena.¹² Therefore, it is indeed quite possible that SNPs A and D from the above example may be more strongly associated with the phenotype and in tighter LD with each other compared to B and C. It has been observed before that such irregularities make it more challenging to localize causal variants, which is why some have suggested focusing on haplotype blocks instead.¹³ To understand whether testing contiguous blocks is indeed a limitation, one must distinguish between fine-mapping based on genotyped SNPs (as it is the case in this paper) and fine-mapping involving all variants (either sequenced or imputed).

In the setting of this paper, where we only analyze genotyped SNPs (Section II C), it is not clear that reporting $\{A, D\}$ as a single discovery would be more informative than reporting $\{A, B, C, D\}$. If we do not have direct information on all variants, we must consider the possibility that the causal variants might be among the untyped ones. Therefore, we have to give $\{A, D\}$ and $\{A, B, C, D\}$ the same interpretation: suggesting the presence of at least one causal variant somewhere within the interval $[A, D]$ or in its immediate vicinity (Section II D 3, and Supplement, Section S4 G 3). From this perspective, $\{A, D\}$ is not more precise than $\{A, B, C, D\}$; in fact, the former can be misleading because the true causal variant has no more reason to be close to A or D than to be anywhere else in the interval $[A, D]$.

If sequencing data is available, it is possible that exclusively testing contiguous groups may not be the best approach for fine-mapping. Our methodology can in principle be applied to whole-genome sequence data, but for this to be practical one must first address additional computational difficulties as well as the intrinsic statistical difficulty of testing conditional hypotheses at such high resolution. Given that the availability of whole-genome sequencing data is still limited, existing methods for fine-mapping often rely on imputation. We will discuss the information contained in imputed variants, and how it can be leveraged in our framework, later in Section R2 B 6.

Further, even if the contiguity requirement were relaxed (...), the need to prespecify groups to be tested may still limit the resolution to which associations can be refined.

Even though we test pre-specified groups at each resolution, which is not the typical fine-mapping approach (as clarified in Section IID3), our method can adapt to the signal strength and refine the position of causal variants by analyzing multiple resolutions, including that of single SNPs, where there is no grouping. Such high-resolution discoveries are difficult to obtain unless the signals are sufficiently strong or the number of observations is very large; however, this is due to a limitation of the data and it does not depend on the groups at lower resolutions. We do not claim that the specific choice of groups applied in this paper is optimal, even though it is informed by the structure of the genetic data. Nonetheless, we have shown that it allows *KnockoffZoom* to localize causal variants powerfully and quite precisely compared to state-of-the-art alternatives.

2. Fine-mapping resolution

For this reason I think it is premature to claim "...KnockoffZoom unifies locus discovery and fine-mapping into a coherent statistical framework" (p15). Specifically, I think its abilities to solve the fine-mapping problem are not yet adequate to make this claim, and that studies interested in fine-mapping will continue to want to use existing Bayesian fine-mapping methods like SUSIE ... In any case, the limited resolution that comes with testing contiguous pre-specified marker groups should be better highlighted in the text.

Following Professor Stephens' suggestion, we have edited the paper to better highlight that our approach is different from that of alternative fine-mapping methods, as we test pre-specified hypotheses (Sections IIA, IID3) and only analyze typed variants (Section IIC). The analysis of imputed variants raises subtle issues that we will discuss in Section R2B6. We compare the performance of our method with that of CAVIAR and SUSIE using genotyped variants in thorough simulations (Section IID3). The results show that *KnockoffZoom* typically performs better than CAVIAR and at least comparably with SUSIE in terms of power and in terms of a principled (although not unique) measure of resolution. We will discuss in the next section a comparison based on alternative measures of resolution.

Regarding the interpretation of the simulation results, it is worth noting that we have given some advantage to the alternative methods by ensuring that their modeling assumptions are satisfied and by carefully applying them "post-BOLT-LMM" in such a way as to avoid selection bias (Supplement, Section S4G), while our method relies on no assumptions whose validity may dif-

fer between the simulated and real data, as discussed in Section IID 1. Along these lines, our method can be applied to binary traits (e.g., Figure 6), and to any other traits that do not follow a homoscedastic Gaussian linear model—the same cannot be said of its alternatives.

Besides better highlighting this limitation in text, the comparisons with fine-mapping methods should be extended to quantify the effect. Currently the comparisons show the “width” of region identified by each method (Figure 4, right panel). However, fine-mapping methods do not strictly identify a region but a set of SNPs, so the figure should also compare the number of SNPs identified by each method.

We have added a new plot in the Supplement (Figure S9) showing the resolution of *KnockoffZoom* and the fine-mapping methods in terms of the average number of SNPs in the discovered groups, along with a reference in Section IID 1 of the revised manuscript. By this metric, and as expected given the contiguity requirement, the resolution of *KnockoffZoom* is lower than that of the other methods but comparable, unless the power of all methods is very low.

It would also be informative to show the minimum pairwise LD between the markers identified – does knockoffZoom sometimes report markers not in high LD with one another due to the contiguity constraint?

Following this suggestion, the new plot mentioned above (Supplement, Figure S9) also compares the resolution of *KnockoffZoom* and the fine-mapping methods in terms of the homogeneity of their discoveries, measured in terms of the average squared correlation within each group (i.e., the average of the squared within-group correlation matrix over all entries). These results show that *KnockoffZoom* sometimes reports together markers that are not in high LD with one another compared to those reported by SUSIE, as expected given the contiguity constraint and now clarified in Section IID 3. However, as the signal strength increases, the impact of the contiguity requirement decreases. When the signal is sufficiently strong, the *KnockoffZoom* discoveries are approximately as homogeneous in terms of LD as those of SUSIE. The *KnockoffZoom* discoveries are usually more homogeneous in this sense than those of CAVIAR, which cannot separate causal variants in one locus into separate findings.

Incidentally, the y axis on this figure is too large to see the interesting region, which for fine-mapping is < 0.1 Mb. Getting to a region of 0.5 Mb is not really fine-mapping in my opinion.

We are grateful to Professor Stephens for pointing this out. We have modified accordingly the

y-axis scales in Figures 4 and S8, which now ranges between 0 and 0.2 Mb (a few points are above 0.1 Mb, so we find this range to be the most informative in this case).

It would also be interesting to get the authors' perspective on how easy or difficult it might be for the contiguity requirement to be relaxed in the future. (Also the pre-specification requirement, although this seems more fundamental.)

As discussed above, the contiguity requirement is easily avoided, but the pre-specification is more fundamental. We view this as an exciting avenue for continued methodological development.

C. Other main comments

1. Consistency of discoveries at different resolutions

Some aspects of Table 1 are surprising to me. Eg the number of bmi findings going from 24 \rightarrow 0 \rightarrow 15 as resolution increases. Shouldn't power increase as larger groups are tested? (I realize there are fewer tests as groups get bigger...so this is not a simple issue.) The hypothyroidism results are perhaps even weirder. Can you provide any intuitive explanation for why this might occur? Is it simply chance, since the knockoff procedure can produce different results if run multiple times?

We thank Professor Stephens for bringing up this point, which needed clarification. It is true that power should increase as larger groups are tested; however, this does not imply that the observed number of discoveries must also increase, for at least two reasons. Firstly, as correctly observed by Professor Stephens, the total number of tests necessarily decreases as more variants are grouped together, and so does the total number of null hypotheses that are not true. Therefore, it is the proportion of rejected false null hypotheses that can be expected to increase, not their total number. Whether their total number increases or decreases depends on other details, such as the number and the spatial distribution of the causal variants. Secondly, power is defined as the *expected* proportion of false null hypotheses that are rejected. The actual observed proportion depends on the data and on other randomized aspects of our procedure, such as the Monte Carlo generation of knockoffs and the computation of test statistics involving cross-validation. We have clarified this in the revised Section IIE 1.

Furthermore, Table I includes some floating discoveries—their presence is particularly clear in the case of hypothyroidism, as correctly pointed out by Professor Stephens. We have not explicitly avoided floating discoveries in the paper (although this is done in the revised supplement, as discussed next) because they are not uninformative. In fact, floating discoveries may indicate

true but weak signals whose significance is close to the FDR threshold and hence unstable to perturbations due to knockoff randomness and cross-validation.

The introduction criticizes the two-step approach as “not fully satisfactory because it requires switching models and assumptions in the middle of the analysis, obfuscating the interpretation of the findings and possibly invalidating type-I error guarantees.” However, from Table 1 (see above comment), performing separate analyses at different resolutions appears to have similar problems regarding interpretation.

The findings obtained at each resolution are our final output. This is a product that satisfies desirable properties: (1) it does not need to be clumped to eliminate redundancy because it reports distinct discoveries, in the sense that each is likely to be conditionally important given all other associations; (2) it is endowed with a clear guarantee of FDR control obtained within a principled and consistent statistical framework. This is what we intend to underscore in the introduction and we stand by this statement. The difficulties that Prof. Stephens highlights are linked to the interpretation of results across resolutions. The theoretical guarantees that we can offer for this are more modest, and we have been careful in stating such limitation in the paper. Based on Prof. Stephens’ input, we have clarified this further in the revised Section IID3. The following points allow us to discuss the matter even more explicitly.

The method that avoids “floating” discoveries at high resolution (Supplement S1B) seems to address this, but at a cost in power. What is that cost in power for the analyses here? How does Table 1 look if you apply that method? ...

Professor Stephens correctly notes that one can overcome some of the difficulties of cross-resolution analysis by applying the variation of *KnockoffZoom* that explicitly avoids floating discoveries. At his suggestion, we have now applied this version and compared the new results with those previously reported in the paper: see Supplement (Sections S4J and S5D), with references in Sections IIA, IID1, and IIE1 of the paper.

The new results show that the coordinated method without the theoretical 1.93 correction factor controls the FDR at the correct level in practice, at the cost of a very small power loss. By contrast, the inclusion of the theoretical 1.93 correction factor appears to be overly conservative and can lead to a significant loss of power (this is consistent with the findings in previous work¹⁴). On balance, keeping the floating discoveries is simpler, more informative, and with precise FDR guarantees: thus, we have preferred to discuss these results in the revised supplement rather than in the main text.

2. Variability upon knockoff resampling

As I understand it the output at each resolution depends on a single generation of the knockoff variables, and so the method will report different significant results each time it is run? Is this correct? If so, how different/similar are the results if you run things a second time with another knockoff realization? (It could suffice to do one trait twice to illustrate this)

Correct, if the method is run using two different sets of knockoffs, it may report two different lists of discoveries, although the FDR is controlled in both, as we have now explicitly discussed in the revised Section III. Even though Professor Stephens did not explicitly mention this idea, his question suggests that it would be interesting if we could combine these results in a principled way through some form of meta-analysis. Unfortunately, we do not know how to do this in a powerful way while controlling the FDR. The main difficulty is that discoveries obtained with the same data but different knockoffs are not independent, as it has been noted before.^{15,16} In truth, this problem goes beyond the scope of *KnockoffZoom*, as it would be at least as challenging to combine different results obtained with any alternative procedures involving either randomness (e.g., data splitting, cross-validation, Monte Carlo sampling, ...) or subjective user choices (e.g., prior specification, choice of test, data pre-processing, ...). In general, this problem seems very challenging from a theoretical perspective, but we suspect that good heuristic solutions could be developed in the future, perhaps just focusing on the special case of knockoffs.

Meanwhile, we agree that it is interesting compare empirically the *KnockoffZoom* results obtained using different sets of knockoffs. For this revision, we have generated a different set of knockoffs and applied them to repeat the analysis of the UK Biobank phenotypes, as discussed in the revised Section III.. The results reported in the revised Supplement, Section S5 J, demonstrate that the findings of *KnockoffZoom* are quite stable to knockoff resampling (between 75% and 85% of them do not change), provided that the number of discoveries is not too small. By contrast, we have observed that if the number of discoveries is of order $1/q$, where q is the nominal FDR level ($q = 0.1$ in our case), the discoveries tend to be less stable, although this also depends on the signal strength. In fact, such variability mostly affects discoveries near the FDR threshold, and about whose individual significance we are less confident; see revised Supplement, Table S18. We refer back to the Supplement, Section S4 K, for more details about the individual significance of each discovery. Finally, we have also verified that the vast majority of the discoveries detectable by BOLT-LMM, even when the latter is applied to a larger sample, are consistently found by *KnockoffZoom* using different sets of knockoffs; see revised Supplement, Table S22.

3. Numerical stability of test statistics

The notation (X, \tilde{X}) suggests that the knockoffs are always included after the real variables in the input file to the lasso/bigsnpr. In principle the location of the knockoffs in the input file should not matter ... As a safety check I suggest running the method with randomly ordered columns, or if that is too much of a pain simply reversing (X, \tilde{X}) to check it makes no difference.

This is an excellent observation. Our software randomly swaps each pair of genotypes and knockoffs before computing the variable importance measures (e.g., by fitting the lasso) to avoid this issue. The original identity of the knockoffs is only revealed later to determine the sign of the test statistics. We have now explicitly discussed this point in the revised Supplement (Section S3 C).

4. References to the implemented HMM

I found the references to the Li-Stephens model vs fastPHASE model in the Supplement confusing. ... Please clarify.

We apologize for this inaccuracy; we have corrected it in the revision.

5. Analytical calculations

The results in the supplement that reduce forward-backward calculations to $O(K)$ and $O(K^2)$ look similar to results that are already well established (e.g. Fearnhead and Donnelly, 2001, Estimating recombination rates from population genetic data, Genetics). Is there anything new here?

The novelty in our analytical results lies in the knockoff generation (Supplement, Section S2 C), which is completely new. The forward-backward calculations used to reconstruct the Markov chain from the HMM (Supplement, Section S2 D) are simpler and indeed related to the earlier work mentioned by Professor Stephens,¹⁷ although we were not aware of this before. This is why we have re-derived them independently instead of citing the appropriate reference. We have added the missing reference in the revised Supplement, Section S2 D 2, but we find it clearer to preserve the exposition in the current form and notation, especially because it is easy to derive the efficient forward-backward algorithm starting from the proof of the knockoff generation results in Section S2 C.

6. Additional software information

Please provide more details about the comparisons with other methods, including versions of software and the settings used. Ideally the code used to run the comparisons with other methods should be made available- even without documentation . . .

We have included this information in the revised Supplement (Section S4L) and mentioned it in the revised paper (Section IID1). We have also shared the code used to run the comparisons in our simulations here: https://github.com/mnesia/ukbiobank_knockoffs.

D. Other comments/questions

1. From array SNPs to full-genome data

Getting the method working on problems of UK biobank scale is impressive, even though limited to “only” 591k SNPs. Would applying to 50 million SNPs be feasible, and require about 100 times the computation? For coarse resolution it might not matter much to include the extra SNPs, but for fine-mapping it ultimately seems important to include as many SNPs as possible.

Applying *KnockoffZoom* on 50 million SNPs is feasible in principle but the current implementation would be expensive in practice. We plan to address this technical limitation in the future, as now discussed in the revised Section III. Constructing knockoffs for 50 million SNPs would require 100 times longer than constructing knockoffs for 500,000 SNPs. However, note that: (1) our algorithm is parallelizable; (2) it may be reasonable to first investigate genome-wide signals on a coarser scale and then construct high-resolution knockoffs only for variants in the regions that show some signal, thus substantially reducing the computational cost. Currently, another problem is that fastPHASE cannot handle the estimation of the HMM parameters for 50 million SNPs. We are working on a new version of *KnockoffZoom* that does not rely on fastPHASE to fit the HMM.

2. Distribution of correlations between knockoffs and genotypes

The paper discards tests where the knockoffs are very highly correlated with the original variables (which makes sense as they have no power). For intuition I would be interested to see the distribution of the correlation of knockoffs with the original variables (say at the finest resolution).

We have provided these plots in the Supplement, Figure S2, with a reference in Section S3C.

3. *Distribution of minor allele frequencies*

What is the MAF distribution of the variants analyzed here? Does the method work equally well for common vs rare variants? (I ask because . . .)

We have summarized the distribution of allele frequencies in our data in the revised Supplement, Table S16. We agree that the LD models that we use to generate knockoffs describe more accurately the distribution of more common variants, as it is now discussed in Section II A. This intuition is confirmed by the exchangeability diagnostics discussed later in Section R2 B 5. This limitation leaves room for further improvements in the future (we are currently working on an improved model for the generation of knockoffs that will also take better care of rare variants), as now mentioned in Section III, but it is not a problem within the scope of this paper. Firstly, *KnockoffZoom* controls the FDR in practice in the simulations, which are based on the real genetic data of the UK Biobank and therefore replicate any possible model misspecification, including the imperfect modeling of the distribution of rarer variants (see Section R3 B 2 for a more detailed discussion of the replication of model misspecification in our simulations). Secondly, the data analysis indicates no bias towards rare variants when *KnockoffZoom* is applied to real phenotypes. On the contrary, Table S16 shows that *KnockoffZoom* mostly selects common variants, which is not surprising since associations on rare variants are generally more difficult to detect. Finally, the new results discussed in the Supplement (Section S4 H) show that *KnockoffZoom* performs well in our simulations even for lower-frequency variants.

4. *Additional references*

It would perhaps be helpful to cite (and contrast with) previous work that attempts to control error rates of conditional tests of groups of variables (eg work on hierarchical testing by Yekutieli, Meinhausen, Bühlmann etc).

The existing statistical literature on hierarchical testing or high-dimensional inference is extremely rich and, having worked in some aspects of it, we are quite fond of it. Still, surveying it is not the objective of this paper, especially because it is largely unrelated to the problem considered here and to the specific practical method we propose (we test conditional hypotheses at each resolution separately from the other resolutions, and make statements only on resolution-specific FDR control). Having said this, we have followed Prof. Stephens' suggestion and added in Section III references to earlier proposed multivariate and hierarchical methods for GWAS.

E. Minor comments

We have made all the necessary corrections in the revised manuscript.

- (i) Page 6: *“by likelihood of the trait”* → *“in distribution of the trait”* ;
- (ii) Page 11: *“its intrinsic limitations discussed above”* - *I do not see where they were discussed.*
 Professor Stephens is right, the sentence in question was lost during the final stages of the manuscript preparation. We have corrected this mistake in the revised manuscript.
- (iii) Page 16: *“possible *to* construct”* ;
- (iv) Page 12: *“As the resolution increases, we report fewer findings”* - *not always!* This issue has already been discussed in depth in Section R1 C 1. We have also clarified it on page 12 of the revised manuscript.
- (v) Table I: *I suggest giving resolution in terms of kb instead of Mb.* We prefer to keep the resolution in Mb because typical values are between 0 and 1 in these units, which is neat, as well as for consistency with other parts of the paper in which Mb are clearly more appropriate (e.g., Figure 1).
- (vi) Table I: *Is 0.000 down to single SNP resolution?* Yes, we have made this clearer now.
- (vii) References: *markov* → *Markov* ; *uk* → *UK* .

R2. REVIEWER 2

A. Overview

This paper describes a new method for jointly identifying and fine-mapping causal variants in genetic association studies. This is a central problem to the field and the very novel approach taken by the authors opens a new avenue for exploration in comparison to the rather similar approaches of existing methods. Therefore, while some of the empirical benefits are modest, the paper is an important addition and will hopefully be expanded in future work.

We sincerely appreciate the positive overall comments of this reviewer, and we beg to disagree that the “empirical benefits” are only modest.

- **Power (locus discovery).** The power of *KnockoffZoom* to discover associated loci in Figure 3 is about twice as large as that of BOLT-LMM throughout most of the heritability range. In fact, *KnockoffZoom* has the same power of an *omniscient FDR-controlling oracle* based on the BOLT-LMM p-values, even though it does not have any of its practically impossible advantages. Moreover, consider the results of the UK Biobank data analysis in Tables I, which show that *KnockoffZoom* discovers many more distinct associations than BOLT-LMM, even when the latter is applied to a much larger sample (Supplement, Table S8), and despite the fact that it is exceedingly generous to consider most of these BOLT-LMM findings as “distinct”, given that they indicate overlapping genetic segments.
- **Power (fine-mapping).** *KnockoffZoom* has approximately three times larger fine-mapping power compared to CAVIAR in Figure 4, while also achieving higher resolution. It is true that in this particular example *KnockoffZoom* does not perform better than SUSIE, but we have given SUSIE some advantages, by ensuring that its strict modelling assumptions are realistic and by carefully clumping the BOLT-LMM discoveries to avoid selection bias (Supplement, Section S4 G). Despite the same advantages, SUSIE does not perform as well as *KnockoffZoom* in terms of power in other simulations (e.g., Supplement, Figure S8) because it is tied to BOLT-LMM, which is suboptimal for locus discovery.
- **Resolution.** The *KnockoffZoom* discoveries in Figure 3 are more precise than those of BOLT-LMM, even when our method is applied at the lowest resolution. The fine-mapping resolution of *KnockoffZoom* in Figure 4 is higher than that of CAVIAR, and not much lower than that of SUSIE when the signal is strong.

- **Flexibility.** *KnockoffZoom* is not limited to traits following a homoscedastic linear model, whose validity would be hard to verify in practice. In fact, our method applies seamlessly regardless of the distribution of the phenotype, which may also binary, as in Figure 6.

B. Comments

There are several additions and clarifications that would be beneficial.

We thank the reviewer for giving us the opportunity to expand the discussion of points below.

1. Contiguous groups

The fine-mapping methods compared to can explore causal variants that are not contained in contiguous groups. For example, CAVIAR explores arbitrary assignments of causal variants across the entire associated locus. The authors should explain this ... in the main text.

We discuss the choice of contiguous groups at length in Section R1 B 1, and we have added a paragraph in Section IID 3 of the revised paper to highlight this point mentioned by the reviewer. Here, we would like to say a little more about CAVIAR. It is now well-known that an important limitation of CAVIAR is that it cannot distinguish between multiple signals in the same locus,¹⁸ as mentioned in Section IID 3 of the paper, and in Section S4 G 1 of the supplement. Therefore, it is not very accurate to say that CAVIAR explores arbitrary assignments of causal variants. Instead, CAVIAR can discard from an associated locus those variants that are likely to be non-causal, under certain precise assumptions. This does not really answer the ultimate question of fine-mapping: which variants *are* likely to be causal? Consider the following example: a promising locus contains four SNPs, $\{A, B, C, D\}$, two of which are causal, namely A and D . The best that CAVIAR can do is to return $\{A, D\}$ as output, but it cannot say whether this indicates that *either* A or D are causal, or *both* are. By contrast, *KnockoffZoom* and SUSIE have the ability to arrive at the correct answer by reporting $\{A\}$ and $\{D\}$ as distinct discoveries.¹⁸

2. Measuring resolution

Along those lines: How is width defined for the results of caviar and other methods? Is it the case the width is smaller for these methods and by how much? Can you also compare on their favored of credible set? That is, can you specify how many additional variants are needed for each of the methods before the causals are included as they do in their papers?

For all methods, the width is measured as the distance in base pairs between the leftmost and rightmost SNPs in each group reported as a distinct discovery. We have clarified this definition in Section IID 1. We have also compared the resolution of the discoveries reported by different methods according to alternative metrics in the revised Supplement (Figure S9), as referenced in Section IID 3; see also our response to the first reviewer in Section R1 B 2.

We have decided against following the reviewer’s last suggestion. This might be a useful comparison for methods that provide fine-mapping analysis of one locus only, when it is sure that it contains causal variants. However, it becomes less meaningful in context like ours, where the entire genome is analyzed with FDR control of discoveries and practitioners commit in advance to a certain significance level.

3. Knockoffs for non-contiguous groups

Is it possible to generate knockoffs from non-contiguous groups to achieve better resolution?

Yes, this is possible by temporarily splitting those groups before generating the knockoffs, as mentioned in the Supplement, Section S2 A 1. We have also clarified in Section II A that contiguity is not required. Our choice of working with contiguous groups in this paper is discussed more thoroughly in Section R1 B 1 of this response.

4. Alternative two-step method

With 350k individuals is it possible to run the desired penalized multivariate regression in the associated locus in many instances? If so, how does this approach perform relative to the others?

We confess having had some trouble in understanding exactly this suggestion. In the following, we have interpreted it as referring to a possible two-step method based on BOLT-LMM (or some other standard locus-discovery method) followed by penalized multivariate regression, instead of the typical fine-mapping tools considered in this paper, such as CAVIAR and SUSIE. Practically, this is conceivable, but we fail to see why it would be desirable (we may perhaps have misunderstood the question, in which case we apologize). Firstly, this hypothetical method would have no statistical guarantees. In this sense, the two-step method considered in this paper is much more principled because CAVIAR and SUSIE at least have some guarantees in the second step, given several assumptions on the locus given by the first step. Secondly, if we were willing to use pe-

nalized multivariate regression without guarantees (which we certainly do not advocate), it would be computationally feasible to apply it genome-wide, so there would no longer be any point in performing the first step of locus discovery with BOLT-LMM.

5. Knockoff diagnostics and rare variants

It is well known that the HMM models of genetic data, including Li & Stephens, are worse for rare variants. Do you see a relative decay in performance at rare variants compared to other methods?

This is a good point to keep in mind, and we thank the reviewer for bringing it up. We have mentioned this limitation in the revised Section II A and provided further details in the Supplement (Sections S4 H and S5 G). There is indeed some decay in the accuracy of the LD model used by *KnockoffZoom* to generate knockoffs (see our answer to the next question below for further details), but this does not seem to be a problem in practice within the scope of this paper (see also our reply to a similar question from the first reviewer in Section R1 D 3). However, this limitation may be problematic in different settings where rarer variants are of greater interest. We are currently implementing a more flexible HMM inspired by the modelling approach of SHAPEIT¹⁹ to generate more accurate knockoffs for lower-frequency variants, as well as to better account for population structure, as discussed in Section III. We will present these results in a later paper.

Can you and have you explicitly tested the knockoffs generated for the properties required? In particular, have you looked at exchangeability of the rare variants?

We are grateful to the reviewer for giving us the opportunity to expand on this point. We now include the goodness-of-fit diagnostics of the HMM with real genotypes (Supplement, Section S4 C; referenced in Section II D 1). In that section, we compare the exchangeability of the knockoffs with the original genotypes in terms of the r^2 measure of LD, while stratifying by minor allele frequency. The results show that our current implementation of the HMM does lead to approximately exchangeable knockoffs, but suffers from some limitations with regard to rare variants and long-range LD (possibly due to some latent population structure). These limitations do not seem to affect the practical performance of *KnockoffZoom* within the scope of this paper, as already discussed in Section R1 D 3, but suggest future work for applications involving data sets with rarer variants or stronger population structure. We are actively working in this direction.

6. Imputed variants

Can the method be adapted to handle imputed data and what happens when the causal variants are in the imputed set? An examination and discussion of imputed data in warranted given its pervasiveness in GWAS.

This is an excellent question that we believe will benefit particularly from a deeper discussion. We have addressed this point in Section IIC of the revised manuscript and in Section S6 A of the revised supplement. In a nutshell, the idea is that we have not included imputed variants in our analysis because they fundamentally contain no information on any phenotype in addition to that carried by the genotyped variants. Even though imputed variants cannot help us increase the resolution of our findings, it may be possible to leverage them to compute more powerful test statistics; this is something that will be interesting to explore in the future.

7. Assessing the individual significance of each discovery

It is worth mentioning section “Assessing the individual significance of each discovery” in the discussion because it is likely to be used in practice.

We are glad to hear that this is of interest and we have followed the suggestion, adding a reference to the supplement in Section IIA and modifying the existing reference in Section III to make it clearer, also in light of the new discussion related to the reviewer’s comment in Section R1 C 2.

8. Towards prediction

It seems likely that removing null regions from a PRS will improve prediction accuracy in a similar manner improving tests of enrichment for causal loci.

We agree that identifying the variants that contribute distinctly to the phenotype (with FDR control) can be the first step towards building more robust predictions with polygenic risk scores. This is mentioned in the last paragraph of Section III. We are aware that a deep connection has been noted before in the statistics literature between multiple testing with FDR control and optimal prediction,^{20–22} even though the existing theory focuses on more or less idealized cases and is not immediately applicable to GWAS data. We may try to extend these ideas and translate them into practical methods for GWAS in the future, as many challenges are open. For instance, it is not yet clear what the FDR level should be if our main focus is to make the accurate predictions in finite samples, or how the resolution level of *KnockoffZoom* should be tuned for this purpose.

C. Minor Comments

- (i) Figure 1: *NO (a) (b) (c) etc* We have fixed this, thank you for bringing it to our attention.
- (ii) Figure 2: *It would be helpful to see the number of false discoveries as a function of h_2 similar to the subfigure in the lower left. Unless these are identical at 10% for all methods?* If we denote the numbers of true and false discoveries as TP and FP, the FDP is defined as $FDP = FP / (TP + FP)$. From this it follows that $FP = FDP \cdot TP / (1 - FDP)$. Therefore, we had to choose which of these redundant quantities would be best to present. We feel that the FDP is the most meaningful measure of type-I errors to plot in this situation. Furthermore, *KnockoffZoom* and the BOLT-LMM oracle are both calibrated to control the (expected) FDP. The practical version of BOLT-LMM (5×10^{-8}) should control the FP and FDP at essentially zero, but it fails in part (b) of Figure 2 because it is testing the wrong hypotheses.
- (iii) Supplement, Section S1 B: *I am not able to review section “Coordinating discoveries across resolutions” because I lack the background in that area of statistics.* We appreciate the reviewer’s candor.

R3. REVIEWER 3

A. Overview

... The work is carefully presented and accompanied by a released software package, but it can be improved in several aspects.

We are grateful for the suggestions below, which we have fully taken into account in the revision.

B. Comments

1. Abstract and introduction

I find the abstract and the introduction overstate the ability of the proposed in terms of identifying the causal variants. The latter statement that “The KnockoZoom discoveries, by accounting for LD, bring us closer to the identification of functional variants” is much more accurate than “KnockoZoom localizes causal variants precisely” in the abstract.

We have modified the abstract as suggested. In the introduction, we have clarified that *KnockoffZoom* addresses the localization of causal variants by testing the conditional association of pre-defined groups at different resolutions.

The disposition of the literature review in introduction can be expanded in that methods to be compared with (e.g. CAVIAR and SUSIE) are discussed explicitly.

We have expanded the description of the goals of CAVIAR and SUSIE and added further references to the existing fine-mapping literature in Section S4 G 1 of the supplement, with pointers in Sections IID 1 and IID 3 of the revised paper. We have also added a new comparison of their outputs with that of *KnockoffZoom* in Section S4 G 3 of the supplement, which is referenced in Section IID 3 of the revised paper. Although more details might make the manuscript even more self-contained, we fear that further expanding the literature review would unnecessarily increase the length and detract from the main message of the paper. We have already given a concise but clear introduction, highlighted the differences between *KnockoffZoom* and the state-of-the-art methods, provided the appropriate references, and thoroughly compared their practical performance in simulations. We anticipate that many of our interested readers will already be more or less familiar with the existing fine-mapping literature.

2. Population structure

The lipid illustrative example is misleading in that it implies that the traditional GWAS does not account for population structure inferred from all genotypes; incorporating PCA is a standard and effective practice that deals with the confounding issue illustrated by the lipid example.

We have followed this suggestion and clarified this point in the revised Section II B. In particular we have clarified that the standard approach does indeed include PCA, or relies on mixed effect models, to handle population structure. Our point in this discussion is to show that the inclusion of PCA is unnecessary in our conditional analysis.

Further, the proposed “inferences rely on a model for the distribution of the genotypes, which requires some approximation. . . . Thus, how the proposed “conditional hypotheses are less susceptible to the confounding effect of population structure” is not clear. The sensitivity of the method to the HMM approximation is also not clear.

Within the relatively homogeneous populations considered in this paper, we have already studied the robustness of *KnockoffZoom* to the HMM approximation in our simulations described in Section II D. Following the reviewer’s suggestion, we have expanded the discussion of this very important point in Sections II B and II D 1 during the revision. Quoting from the paper: “we are not protecting *KnockoffZoom* from model misspecification because we use real genotypes and approximate their distribution (our sole assumption is that we can do this accurately) with the same HMM (Supplement, Section S4 A) as in our data analysis.” We further expand on this point here.

Consider a method whose inference rely on parametric modelling assumptions for the conditional distribution of the phenotype Y given the genotypes X (e.g, the Bayesian linear models of BOLT-LMM, CAVIAR, or SUSIE). Since these assumptions are placed directly on the object of inference (i.e., the conditional distribution of $Y | X$), it is difficult to validate the method using numerical experiments. In particular, it would not be fully satisfactory to confirm that it works well with simulated phenotypes Y^* obtained from a linear model with homoscedastic Gaussian errors.

If we wanted to perform the numerical experiments in this paper giving *KnockoffZoom* the same level of advantage we have given to BOLT-LMM, CAVIAR, or SUSIE, we would have created synthetic genotypes from some arbitrary HMM, used them to generate a synthetic phenotype, constructed exact knockoffs using the true HMM, and showed that we control the FDR. However, such experiment would not be very interesting because we can already mathematically prove the

same result; moreover, it would not be informative regarding our sensitivity to population structure, as correctly pointed out by the reviewer. Instead, our numerical experiments are based on the *real genotypes* in the UK Biobank, whose distribution we have approximated with the same HMM used for the analysis of the real phenotypes. The only difference between the simulations and the real data analysis is the conditional distribution of $Y | X$, about which our method makes no assumptions. Therefore, our simulations already explicitly demonstrate the robustness of *KnockoffZoom* to model misspecification, including the possible presence of population structure.

It would go beyond the scope of this paper to explore the robustness of *KnockoffZoom* when the data contain stronger population structure, but we carefully explain in Section IIB that the current implementation of the HMM may not work well in such settings. We are currently working to address the broader question of population structure as part of our continuing work, by implementing a more flexible HMM inspired by the modelling approach of SHAPEIT.¹⁹

3. Missing annotations

Figure 1: (a)-(c) annotation is missing from the plot.

We have fixed this, as also mentioned in Section R2C.

4. Additional explanations for the results of the simulations

There are some features in Figure 3 require explanations, e.g. why the width of the oracle procedure is not monotone as h^2 increases? Why the true discoveries of the oracle procedure are substantially less than the proposed method as h^2 increases in (b)? And why the FDP of LMM increases as h^2 increases in (b)?

We have added the missing explanations in the revised manuscript (Section IID 2), as suggested. We are grateful to the reviewer for bringing it to our attention.

5. Performance comparison with BOLT-LMM + SUSIE

Results in Figure 4 suggest that SUSIE is a better method than the proposed one with lower FDP and narrower width of discoveries. I suggest highlighting their differences in the main text instead of the Supplementary materials.

We understand that one simulation scenario is not sufficient to fully appreciate the relative strengths and weaknesses of alternative methods, which is why we have included additional figures

in the Supplement (e.g., Section S4 G). However, it would not be practically feasible to include all plots in the paper, nor would we want to cherry-pick those that place *KnockoffZoom* in the most favorable light. Moreover, we find it perfectly acceptable that here *KnockoffZoom* performs similarly to the two-step procedure involving BOLT-LMM and SUSIE, even though in other settings the latter may have lower power (e.g., Supplement, Figure S8), suffer from selection bias (e.g., Supplement, Figure S7) or invalid locus discovery (e.g., Figure 6). After all, our numerical experiments have already given some advantages the alternative methods, by simulating a continuous trait from a Gaussian linear model and by ensuring that the clumping and filtering heuristics do not invalidate their inferences, which would not be possible in practice, as discussed in the revised Section IID 1. By contrast, *KnockoffZoom* is applied out-of-the-box and has no unfair advantage in these simulations compared to real data, as already discussed earlier in Section R3 B 2.

6. *Applying fine-mapping tools in data analysis*

The analysis presented in Section IIE should include SUSIE in parallel with the simulation studies in Section IID. Figure 5 should include results from the standard analyses in parallel with the earlier figures.

We understand this comment and we have given it careful thought. However, we feel that it would be confusing and uninformative to apply alternative fine-mapping methods in the real data examples because the ground truth is unknown and the validity of the modeling assumptions of CAVIAR and SUSIE are unverifiable (these assumptions are certainly not valid for binary traits). Moreover, we cannot ensure that the BOLT-LMM results are clumped correctly (Figure 3 b) and two-step fine-mapping procedure avoids selection bias (Supplement, Figure S7). Therefore, it may give a false sense of confidence to confirm our results with the alternative fine-mapping procedures, while almost certainly distracting from the main purpose of Section IIE, which is to show an application of *KnockoffZoom*.

7. *Controlling the FDR*

Finally, the authors discussed extensively their choice of controlling FDR instead of FWER in the discussion section. Still, I find the choice a bit odd given the nature of the study: identifying causal variants as opposed to exploratory analyses. This difference also complicates the later comparison between KnockoffZoom and BOLT-LMM.

The analysis of GWAS data is intrinsically exploratory: follow-up sequencing and functional studies will be needed to establish exactly which genetic variants have a biological effect on the trait of interest. In this context, we aim to control the FDR of distinct discoveries, where each discovery corresponds to a separate possible genetic influence on the trait. We study complex traits, where we expect there to be many such discoveries, so that it does not make sense to penalize a method if it makes a single false discovery out of hundreds or thousands. Knockoffs could be used to control FWER,²³ but this is not appropriate for the problem considered here.

KnockoffZoom aims to localize causal variants genome-wide. All discoveries at one level of resolution are distinct from one another. Depending on the level of resolution, each discovery corresponds to a single variant (which can be interpreted as a causal variant or the proxy to a causal variant) or to genomic segment. BOLT-LMM, instead, returns discoveries at the level of SNPs, but these discoveries cannot be interpreted as distinct and they are indeed not typically discussed as such in publication reports; instead, such discoveries are typically post-processed.

We agree that ideally a comparison between *KnockoffZoom* and BOLT-LMM would involve the same global error criteria, but it is not really possible to control the FDR of distinct discoveries with a standard application of typical FDR-controlling methods to the BOLT-LMM p-values. We have illustrated this issue in Section S4E of the supplement (referenced in Section IID 2 of the paper) and, to remedy to this difficulty, we have used an oracle procedure in the simulations. Unfortunately, in the case of real data analysis we are forced to use the standard pipeline for BOLT-LMM, which aims to control FWER. While this compounds the effects of defining different null hypotheses with that of controlling a different error rate, it has the advantage of offering a head-to-head comparison of what can be considered the current standard analysis and the new one we propose.

Indeed, despite the effort of presenting an oracle procedure the results presented in Figure 3 can be misleading given the differential FDP. I do not request fundamentally changing the error control procedure, but I appreciate more insightful comments on this issue.

While we believe that Figure 3 is self-contained, the reviewer (and the reader) might find it useful to interpret it in light of the data in Figure S5 in the supplement (also discussed in Section IID 2 of the paper), which illustrate the difficulties in controlling FDR with BOLT-LMM. Furthermore, Figure 3 illustrates the effect of the discrepancy between the marginal hypotheses tested by BOLT-LMM and the “conditional” discoveries in which one is interested when looking for causal variants. Even though the LMM p-values are calibrated to control the FWER for hypotheses of *marginal*

association, there is no reason why the final output obtained with LD-clumping heuristics should control the FWER, or even the FDR, for the more interesting hypotheses of conditional association (i.e., distinct discoveries). In fact, Figure 3 b demonstrates that this is not the case: without the oracle BOLT-LMM does not control either error rate (unless the clumping is very aggressive, as in Figure 3 a). Therefore, at worst our comparison is unfair towards *KnockoffZoom*, since we compare it against an oracle that would be impossible to implement in practice.

-
- ¹ Daly, M. J., Rioux, J. D., Schaffner, S. F., Hudson, T. J. & Lander, E. S. High-resolution haplotype structure in the human genome. *Nature genetics* **29**, 229 (2001).
 - ² Gabriel, S. B. *et al.* The structure of haplotype blocks in the human genome. *Science* **296**, 2225–2229 (2002).
 - ³ Zhang, K., Calabrese, P., Nordborg, M. & Sun, F. Haplotype block structure and its applications to association studies: power and study designs. *The American Journal of Human Genetics* **71**, 1386–1394 (2002).
 - ⁴ Wall, J. D. & Pritchard, J. K. Haplotype blocks and linkage disequilibrium in the human genome. *Nature Reviews Genetics* **4**, 587 (2003).
 - ⁵ Beyene, J., Tritchler, D., Asimit, J. L. & Hamid, J. S. Gene- or region-based analysis of genome-wide association studies. *Genetic epidemiology* **33**, S105–S110 (2009).
 - ⁶ Buil, A. *et al.* A new gene-based association test for genome-wide association studies. In *BMC proceedings*, vol. 3, S130 (BioMed Central, 2009).
 - ⁷ Lo, S.-H., Chernoff, H., Cong, L., Ding, Y. & Zheng, T. Discovering interactions among BRCA1 and other candidate genes associated with sporadic breast cancer. *Proceedings of the National Academy of Sciences* **105**, 12387–12392 (2008).
 - ⁸ Qiao, B. *et al.* Genome-wide gene-based analysis of rheumatoid arthritis-associated interaction with PTPN22 and HLA-DRB1. In *BMC proceedings*, vol. 3, S132 (BioMed Central, 2009).
 - ⁹ Hedrick, P. W. Gametic disequilibrium measures: proceed with caution. *Genetics* **117**, 331–341 (1987).
 - ¹⁰ Lewontin, R. On measures of gametic disequilibrium. *Genetics* **120**, 849–852 (1988).
 - ¹¹ VanLiere, J. M. & Rosenberg, N. A. Mathematical properties of the r^2 measure of linkage disequilibrium. *Theoretical population biology* **74**, 130–137 (2008).
 - ¹² Nordborg, M. & Tavaré, S. Linkage disequilibrium: what history has to tell us. *TRENDS in Genetics* **18**, 83–90 (2002).
 - ¹³ Conti, D. V. & Witte, J. S. Hierarchical modeling of linkage disequilibrium: genetic structure and spatial relations. *The American Journal of Human Genetics* **72**, 351–363 (2003).
 - ¹⁴ Katsevich, E. & Sabatti, C. Multilayer knockoff filter: controlled variable selection at multiple resolutions. *Ann. Appl. Stat.* **13**, 1–33 (2019).

- ¹⁵ Candès, E. J., Fan, Y., Janson, L. & Lv, J. Panning for gold: Model-x knockoffs for high-dimensional controlled variable selection. *J. R. Stat. Soc. B.* **80**, 551–577 (2018).
- ¹⁶ Sesia, M., Sabatti, C. & Candès, E. J. Gene hunting with hidden Markov model knockoffs. *Biometrika* **106**, 1–18 (2019).
- ¹⁷ Fearnhead, P. & Donnelly, P. Estimating recombination rates from population genetic data. *Genetics* **159**, 1299–1318 (2001).
- ¹⁸ Wang, G., Sarkar, A. K., Carbonetto, P. & Stephens, M. A simple new approach to variable selection in regression, with application to genetic fine-mapping. *bioRxiv* (2018).
- ¹⁹ O’Connell, J. *et al.* Haplotype estimation for biobank scale datasets. *Nat. Genet.* **48**, 817–820 (2016).
- ²⁰ Abramovich, F. & Benjamini, Y. Adaptive thresholding of wavelet coefficients. *Computational Statistics & Data Analysis* **22**, 351–361 (1996).
- ²¹ Abramovich, F., Benjamini, Y., Donoho, D. L. & Johnstone, I. M. Adapting to unknown sparsity by controlling the false discovery rate. *The Annals of Statistics* **34**, 584–653 (2006).
- ²² Su, W. & Candès, E. J. SLOPE is adaptive to unknown sparsity and asymptotically minimax. *The Annals of Statistics* **44**, 1038–1068 (2016).
- ²³ Janson, L., Su, W. *et al.* Familywise error rate control via knockoffs. *Electronic Journal of Statistics* **10**, 960–975 (2016).

Reviewers' Comments:

Reviewer #2:

Remarks to the Author:

The authors have addressed all of my concerns.

Reviewer #3:

Remarks to the Author:

The authors did a good job addressing my previous comments.

Multi-resolution localization of causal variants across the genome
Reply to the requests of editorial changes

Matteo Sesia, Eugene Katsevich, Stephen Bates, Emmanuel Candès, Chiara Sabatti
Stanford University, Department of Statistics, Stanford, CA 94305, USA

A. Title page

- We have edited the abstract as suggested. The new abstract is less than 150 word long.

B. Main text

- We have shortened the main text, which is now less than 5000 word long.
- We have rearranged the introduction as suggested.
- All current work in the introduction is discussed in the present tense.
- We have removed the numbering from all headings as suggested.
- We have removed the secondary level of subheadings in the Results section.

C. Language and style

- We have made sure that the mathematical terms follow the guidelines throughout the main text and the supplement. In particular, we have correctly typeset vectors in bold.
- Whenever p-values are stated in figures and legends, it is specified that they refer to the BOLT-LMM tests, which are defined in the main text and references. Our method (*Knock-offZoom*) does not use p-values.

D. Methods and data

- We have renamed the Methods section “Methods”.
- We did not work with human participants. We simply analyzed the UK Biobank data set as provided by the UK Biobank. This information, along with our application number, is reported in the Data Availability section.
- We have deposited all of our code in a persistent repository, and provided the relevant DOIs in the Code Availability section in square brackets.

E. End notes

- We have added the sentence “The authors declare no competing interests.” to the Authors Contribution section.
- We have verified that the references are in the correct format, after a few corrections.
- We have updated the references to a couple of preprints that were recently published.

F. Display items

- Our manuscript and supplement do not contain any third-party images.
- All figure legend titles are in the correct format and brief.
- Data in tables have the correct format.
- All figures are in the correct order.

G. Supplementary information

- We have renamed the supplementary sections as requested.
- We have appropriately corrected all references to the supplement.
- We have corrected the numbering for the supplementary figures and tables.
- We have rearranged the supplementary figures and tables so that they appear in the same order as they are referenced in the main text. All supplementary figures and tables are referenced in the main text.
- The cover page of the supplement contains title and author information.